# Meningococci drive host membrane tubulation to recruit their signaling receptors

Audrey Laurent-Granger [1,12], Kévin Sollier[1,2,3,12], Bruno Saubamea [4], Virginie Mignon[4], Nicolas Goudin [5], Yaëlle Wormser [1,10], Morgane Wuckelt[1], Mahmoud Rifai [1], Thomas Heng[1], Lya L'hermitte[1], Marta Conflitti [6], Julie Meyer [1,11], Hervé Lecuyer[7], Anne Jamet [1,7], Nicolas Borghi [3], Philippe Girard[3], Emmanuelle Bille [1,7], Grégory Lavieu [8], Eric Rubinstein [9], Stefano Marullo [2] ✉ & Mathieu Coureuil [1] ✉

Once passed into the bloodstream, bacterial pathogens have a limited time to interact with permissive receptors at the surface of host cells. *Neisseria meningitidis* has developed an extremely effective strategy allowing it to find its receptors in a few seconds. Here, we report that *N. meningitidis* type IV pili exploit the physical properties of host cells' plasma membranes to promote the formation of early tubular membrane structures essential for initial bacterial adhesion. These tubular structures, which form before any signaling events in host cells, concentrate and trap multiple plasma membrane-associated proteins in the vicinity of bacteria, thereby facilitating the selection, interaction and activation of specific adhesion and signaling receptors by bacterial ligands present on type IV pili. Our results define an additional paradigm for the recruitment of specific receptors by pathogenic bacteria, which depends on the physical property of bacterial pili to induce the formation of tubular plasma membrane structures enriched in integral plasma membrane receptors.

Signal transduction in pathogen-infected cells constitutes a pivotal event in the molecular pathogenesis of infection, proceeding as a cascade of biochemical reactions initiated by the interaction of microbial ligands with host cell receptors. Typically, soluble ligands within biological fluids exist in vast molar excess relative to their matching plasma membrane receptors, thereby conferring a high statistical likelihood of receptor engagement and subsequent signaling activation. In contrast, numerous bacterial ligands are structurally anchored to microbial surfaces, such as outer membranes, cell walls, or appendages, which substantially reduces their effective concentration compared to freely diffusible ligands. This spatial constraint markedly diminishes the probability of successful ligand–receptor

[1]Université Paris Cité, INSERM U1151, CNRS UMR8253, Institut Necker-Enfants Malades, Paris, France. [2]Université Paris Cité, INSERM U1016, CNRS UMR8104, Institut Cochin, Paris, France. [3]Université Paris Cité, CNRS, Institut Jacques Monod, Paris, France. [4]Plateforme d'Imagerie Cellulaire et MOléculaire PICMO, Université Paris Cité, US25 Inserm, UAR3612 CNRS, Faculté de Pharmacie de Paris, Paris, France. [5]Platform for Image Analysis Center, SFR Necker, INSERM US 24, Paris, France. [6]University of Padova, Padova, Italy. [7]Laboratoire de microbiologie clinique, AP-HP, Hôpital Necker Enfants Malades, Malades, France. [8]Université Paris Cité, INSERM U1334, CNRS UMR8175, Department of Biomedical and Fundamental Sciences, Campus Saints-Germain-des-Prés, Paris, France. [9]Sorbonne Université, Inserm, CNRS, Centre d'Immunologie et des Maladies Infectieuses, CIMI-Paris, Paris, France. [10]Present address: Institut Pasteur, Université Paris Cité, CNRS UMR 3528, Bacterial Cell Cycle Mechanisms Unit, Paris, France. [11]Present address: Université Paris Cité, Université Sorbonne Paris Nord, Inserm, IAME, Paris, France. [12]These authors contributed equally: Audrey Laurent-Granger, Kévin Sollier. ✉e-mail: stefano.marullo@inserm.fr; mathieu.coureuil@inserm.fr

encounters, particularly under conditions of low receptor density at the host cell surface.

The challenge of receptor localization is especially consequential for bacterial pathogens, for which efficient engagement of host receptors is crucial for colonization, invasion, and persistence. Taking *Listeria monocytogenes* as a model, the inherently low probability of stable ligand–receptor interactions with host tissues is compensated by high inoculum requirements: experimental models indicate that 10-100 million bacteria are typically necessary to elicit intestinal infection[1] or neuroinvasion, whereas as few as 10,000 organisms suffice to establish infection following intravenous administration[2]. Importantly, bacteria circulating in the bloodstream are subject to temporal constraints, as they must bind to and infect target tissues within a limited time frame before elimination by host immune mechanisms[3].

Intriguingly, *Neisseria meningitidis* exhibits a poorly understood facilitative mechanism whereby as few as five diplococci are sufficient to induce septicemia in murine models, highlighting its exceptional invasive potential[4]. This remarkably low infectious dose is notable given the organism's inability to survive immune cell-mediated clearance checkpoints in organs such as the liver and bladder[4]. For successful colonization of the vascular endothelium, *N. meningitidis* must establish stable adhesion with endothelial cells within a transient window of mere seconds, a process rendered even more difficult by the persistent shear stress imposed by bloodstream flow[5]. Bacterial adhesion is mediated by type IV pili (T4P), which trigger extensive remodeling of the endothelial plasma membrane, resulting in the emergence of elongated tubular, villus-like projections[6–10], hereafter referred to as "tubular membrane structures" (TMS). The biogenesis of these TMS, which are indispensable for tissue colonization, and their function during the initial phases of vascular adhesion, remain largely unresolved.

T4P pili are filamentous organelles widely distributed among monoderm and diderm bacteria as well as archaea[11]. These dynamic polymers play pivotal roles in aggregation, biofilm formation, DNA uptake, mechano-sensing, adhesion or interaction with biotic and abiotic surfaces, features that are essential for bacterial virulence and pathogenesis[12–14]. In many clinically relevant human pathogens, including *P. aeruginosa*[15], *N. meningitidis*[16,17], *N. gonorrhoeae*[6], *Clostridium difficile*[18], enteropathogenic and enterotoxigenic *E. coli* (EPEC and ETEC)[19–21], and *Streptococcus sanguinis*[22], T4P mediate adherence to host cell receptors and initiate signaling cascades that modulate host cell responses. Despite their central role in host–pathogen interactions, relatively few investigations have dissected the molecular mechanisms underlying T4P-receptor engagement or defined specific molecular targets. Available evidence predominantly highlights the capacity of T4P to recognize and bind glycans, either displayed on host cell surface receptors[9,15,23,24] or present within mucins constituting the epithelial mucus barrier[25].

T4P are dynamic structures whose biogenesis depends on sophisticated multi-protein machinery. Although nomenclature varies among bacterial species[26,27], *N. meningitidis* T4P assembly involves approximately 16 widely conserved essential proteins, including PilC1 and PilC2, PilD, PilE, PilF, PilG, PilH, PilI, PilJ, PilK, PilM, PilN, PilO, PilP, PilQ, and PilW. Within this repertoire, PilE constitutes the major pilin subunit forming the pilus fiber, while PilC1 and PilC2 serve as pilus-tip adhesins[28,29]; notably, only PilC1 mediates adhesion to endothelial cells, as demonstrated by PilC1-null mutants expressing PilC2 that remain piliated but non-adhesive[30,31].

Pili dynamics, namely continuous extension and retraction processes, are driven by the ATPases PilF and PilT, respectively[12–14]. Additionally, the minor pilin PilV is thought to localize along the pilus fiber and contributes to adhesion and signaling in host cells[9,10]. The stochastic cycles of elongation and retraction occur on a timescale of seconds, generating mechanical forces ranging from 50 to 100 piconewtons (pN) per pilus filament, which can sum to approximately 1 nanonewton for bundles comprising 8 to 10 pili[32,33]. These forces exceed the typical rupture forces of protein–protein or protein–glycan bonds, generally in the range of a few tens to a hundred pN[34,35], implying that T4P retraction forces alone could disrupt receptor interactions rapidly. Accordingly, durable bacterial adhesion to host cells likely necessitates additional host-derived stabilizing mechanisms. One proposed example is the exploitation by *N. meningitidis* of the "one-dimensional" wetting properties of the host plasma membrane to form TMS enveloping T4P, thereby reinforcing adhesion under shear stress in the bloodstream[36,37].

Wetting is a mechanical phenomenon describing the capacity of a liquid to spread across a solid surface. Expansion of the contact interface is driven by diverse molecular interactions, including ionic, electrostatic, polar, and van der Waals forces, and proceeds as long as the process remains energetically favorable. This principle underlies phenomena such as capillary action. In the one-dimensional (1D) wetting model proposed by Charles-Orszag and colleagues, the plasma membrane behaves as a fluid, spreading along adherent nanofibers[36].

In *N. meningitidis*, stable endothelial adhesion mediated by type IV pili (T4P) depends on recognition of a host receptor complex comprising CD147 and the β2-adrenergic receptor (β2AR), cross-linked through the cytoskeletal protein α-actinin-4[4,6,7,9,38,39]. Following bacterial attachment, CD147 and β2AR accumulate within TMS, where their sialylated glycans engage the secondary adhesin PilV. Retraction forces generated by T4P are implicated in the mechanical activation of these receptor complexes[9], subsequently triggering actin polymerization thought to stabilize the TMS anchoring sites[37].

Both PilC1 and PilV are essential for initiating adhesion: initial contact involves the pilus-tip adhesin PilC1 engaging an as-yet unidentified host receptor, while PilV reinforces binding via interactions with CD147/β2AR complexes. A unifying model integrating these observations posits that PilC1-induced membrane wetting facilitates local enrichment of CD147/β2AR/α-actinin-4 complexes in proximity to bacterial pili, thereby promoting PilV–receptor engagement and initiating host cell signaling responses.

In this study, we examined the earliest stages of T4P engagement with endothelial cells and found that the initial T4P-induced formation of TMS arises solely from physical cues, independent of host cell cytoskeletal remodeling or intracellular signaling. These membrane protrusions, promoted by a wetting-driven process, non-specifically enrich a broad range of plasma membrane–associated proteins, including those directly involved in meningococcal-triggered signaling. The elevated local concentration of these proteins within TMS enhances the likelihood of productive interactions with T4P ligands, thereby facilitating the activation of downstream signaling cascades. These cascades, in turn, promote the development of larger, cytoskeleton-reinforced protrusions required for stable bacterial colonization. Thus, membrane wetting, by simultaneously expanding the contact interface and concentrating specific receptor molecules, emerges as a general biophysical mechanism that promotes encounters between membrane-associated ligands and their cognate receptors during cell–cell interactions.

## Results

### *N. meningitidis*-induced tubular membrane structures are enriched in plasma membrane proteins

*N. meningitidis* induces the formation of TMS in human endothelial cells, which are necessary for efficient bacterial colonization[5,7,40]. These structures have been considered as the result of signaling pathways elicited in host cells by the pathogen and involving ezrin phosphorylation and actin polymerization[37,41–45]. Here, we investigated the key steps of formation of these structures by examining the distribution of a marker, the endogenous membrane curvature-sensitive CD9 tetraspanin[46], in human umbilical vein endothelial (EA.hy926) cells infected with wild-type meningococci (2C4.3 strain) in vitro. CD9

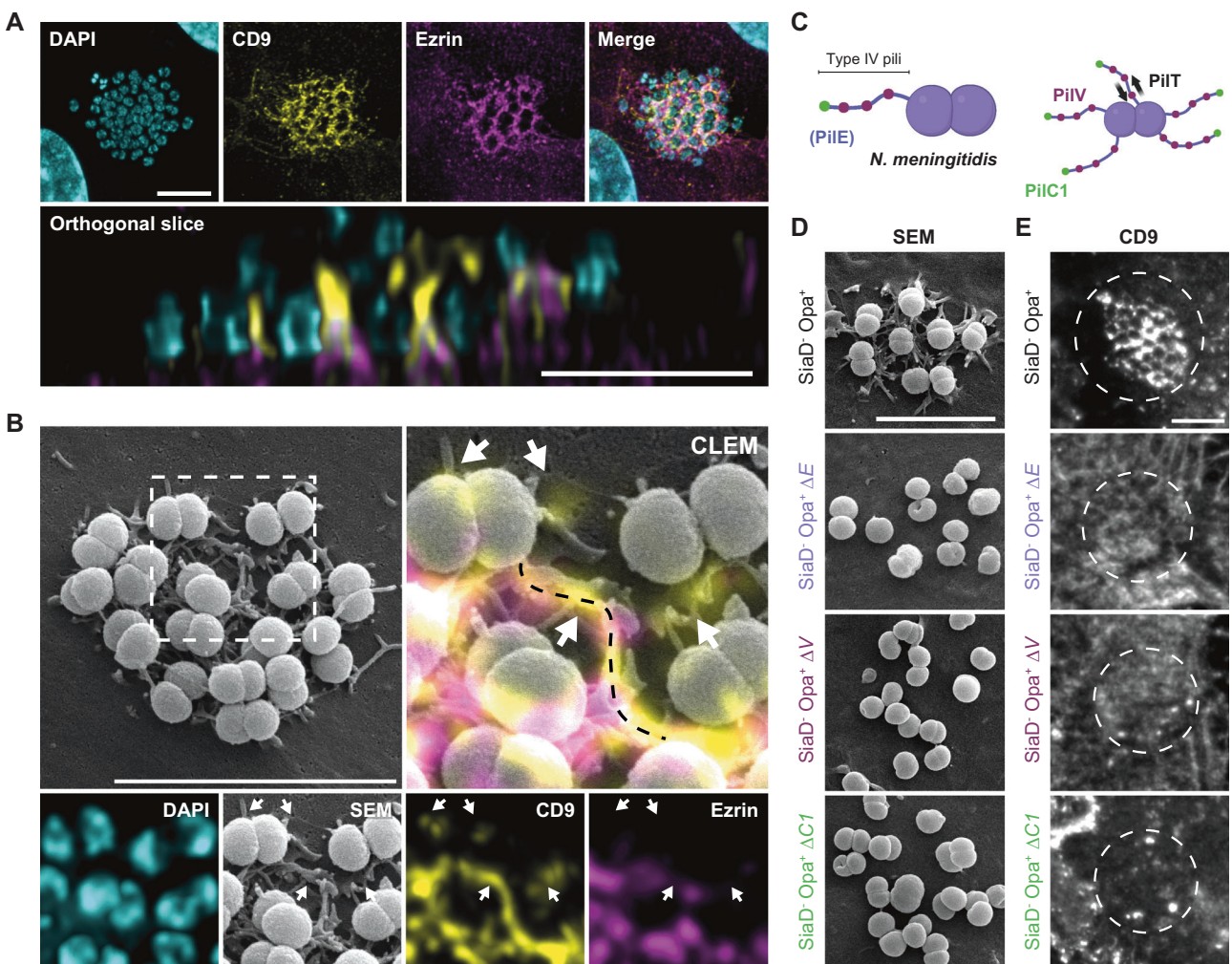

**Fig. 1 | The membrane marker CD9 accumulates in the tubular plasma membrane structures induced by *N. meningitidis*, in contrast to the signaling marker ezrin. A** Representative confocal microscopy images with deconvolution of a wild type meningococcal colony adhering to human endothelial cell EA.hy926. CD9 (membrane marker, yellow) and ezrin (signaling marker, magenta) were immunostained and DAPI was used to reveal cell and bacterial DNA (cyan). Representative of more than 30 cells from two independent experiments. On the top, split and merged channel. On the bottom, orthogonal XZ slice. **B** Representative correlative light and electron microscopy images obtained after repositioning confocal imaging with deconvolution with scanning electron microscopy (SEM) using Icy software and EC-CLEM plugin. Representative of more than 20 cells from two independent experiments. CD9 and ezrin were immunostained and DAPI was used to reveal cell and bacterial DNA. White arrows and dashed black line indicate locations with a clear overlap between CD9 staining and TMS where ezrin staining is absent. (**C**) Schematics of *N. meningitidis* type IV pili (T4P), showing the main pilin PilE (blue), the adhesins PilC (green) and PilV (purple), and the retraction motor PilT (black). Created in BioRender. Coureuil, M. (2025) https://BioRender.com/yxppf77. **D, E** Representative images (three independent experiments) of endothelial cells and colonies of adherent non-encapsulated meningococcal expressing OpaB (*SiaD⁻ Opa⁺*) or the derivative deletion mutants for *pilE* (ΔE), *pilV* (ΔV) and *pilC1* (ΔC1). **D** Representative SEM images. **E** Representative immunofluorescence microscopy images of CD9. Bacterial colonies are indicated by white dashed circles. Z-stack sum projection. Scale bars 5 µm.

labeling near bacterial colonies was reminiscent of that observed with the cell-signaling marker ezrin, which accumulates under *N. meningitidis* colonies (Fig. 1A). However, confocal imaging deconvolution revealed a very poor overlap between ezrin and CD9 labeling, the latter being concentrated in thin TMS surrounding adherent bacteria (Fig. 1A and Supplementary Fig. 1A), which are distant or located above ezrin-positive areas (Fig. 1A, bottom). We then compared CD9 and ezrin distribution using Correlated Light (confocal imaging) and (scanning) Electron Microscopy (CLEM) (Fig. 1B and Supplementary Fig. 1B). While CD9 accumulated into TMS, ezrin appeared to be principally contained into distinct spotty areas with poor colocalization with CD9. TMS were absent at the surface of non-infected endothelial cells (Supplementary Fig. 1C).

Dense CD9 accumulation around meningococcal colonies has been considered as a facilitator of meningococcal adhesion to epithelial cell, reflecting the accumulation of receptors contributing to adhesion[47]. We therefore investigated whether CD9 and other tetraspanins would play an active role in the formation of TMS. We examined the localization of three plasma membrane tetraspanins, CD9, CD81, and CD151 in different types of endothelial cells (EA.hy926, hCMEC/D3 and primary HDBEC) infected with wild-type meningococci (Supplementary Fig. 2A-C). As for CD9 in EA.hy926 cells, all three tetraspanins accumulated around *N. meningitidis* colonies, independently of the endothelial cell type. CD9 knockdown (KD) in EA.hy926 cells had no effect on *N. meningitidis*-induced signaling, assessed by ezrin accumulation (Supplementary Fig. 3A, D) and the same observation was made after silencing CD81 (Supplementary Fig. 3B, E) and CD151 (Supplementary Fig. 3C, F). Consistently, *N. meningitidis*-induced ezrin accumulation was not decreased by the CRISPR-mediated triple knockout (KO) of CD9, CD81 and CD151 in EA.hy926 cells (Supplementary Fig. 3G, H), and the adhesion of wild-type bacteria on these triple KO cells was similar to that of wild-type cells

(Supplementary Fig. 3I). Overall, these data indicate that CD9, CD81 and CD151 tetraspanins were not required for *N. meningitidis* adhesion or for *N. meningitidis*-induced signaling during endothelial cell infection. Several other plasma membrane proteins were recruited to bacterial colonies similarly to CD9: endogenous CD44, some exogenously expressed yellow fluorescent protein (YFP)-tagged G protein-coupled receptors (CXCR4, CCR5, and AT1R), the YFP-tagged T lymphocyte cell marker CD4, and the YFP-tagged chemokine scavenger ACKR2 (Supplementary Fig. 4). Instead, other exogenously expressed plasma membrane receptors, such as the GFP-tagged transferrin receptor or the GFP-tagged Toll like receptors TLR2 and TLR4 were not recruited into TMS (Supplementary Fig. 4).

Bacterial pili are indispensable appendages for in vivo adhesion[8,48] and signaling in host cells[6], while the outer membrane Opacity proteins (Opa) are thought to be involved in later interactions with host cells and, more specifically, with the CEACAM receptors family[49,50]. The main meningococcal T4P structural pilin PilE and the adhesive pilins PilV and PilC1 (Fig. 1C) all contribute to adhesion and signaling (including ezrin accumulation) in endothelial cells. Therefore, to examine the specific functional role of each of these pilins in TMS formation without impeding adhesion, we took advantage of the adhesive function of Opa to prime pilin-independent bacterial adhesion on CEACAM-1 expressing cells[7,44]. This approach requires exposing Opa by inhibiting the formation of the bacterial capsule through mutation of the *siaD* gene[7,44]. The non-encapsulated *N. meningitidis* strain expressing exposed Opa adhesins (SiaD⁻ Opa⁺) still requires T4P to induce the formation of TMS and signaling. The *SiaD⁻ Opa⁺* meningococcal strains consistently promoted CD9 accumulation into the TMS surrounding adherent bacteria (Fig. 1D, E top panels and Supplementary Fig. 2) and ezrin accumulation under colonies[44]. In contrast, the non-piliated derivative ΔPilE mutant and the adhesion-defective derivatives ΔPilV and ΔPilC1 mutant strains adhered to endothelial cells but failed to induce the formation of CD9-positive TMS (Fig. 1D, E and Supplementary Fig. 2). These data indicate that adhesive T4P are necessary and sufficient to generate CD9-positive TMS in endothelial cells.

We then investigated the diffusion dynamics of plasma membrane proteins within the TMS induced by meningococcal T4P. Fluorescence recovery after photobleaching (FRAP) experiments were conducted in endothelial cells expressing YFP-CD9, infected with the 2C4.3 strain (Fig. 2A). YFP-CD9 mobility was significantly impaired in TMS compared to control plasma membrane areas distant from bacterial colonies on the same coverslip (Fig. 2B). Mobility of both the *N. meningitidis* signaling receptor β2AR (β2AR-YFP) and of the T lymphocyte CD4 surface glycoprotein (CD4-YFP), which is not involved in signaling, was similarly reduced in TMS, compared to control distant plasma membrane areas (Fig. 2C, D). These findings suggest that TMS indiscriminately trap cell surface proteins and receptors, regardless of their potential role in *N. meningitidis*-induced signaling in host cells. Such a phenomenon might drive the non-selective accumulation of multiple plasma-membrane-associated proteins in TMS, including the adhesion and signaling receptors for meningococci. To quantify *N. meningitidis* signaling receptor β2AR accumulation into TMS, the fluorescent signal of β2AR-YFP and that of YFP-CD9 and CD4-YFP – both used here as positive controls – was measured in *N. meningitidis*-induced TMS relative to that measured in ordinary filopodia present in areas of the plasma membrane distant from meningococcal colonies (Fig. 2E, F). We reasoned that if membrane shape was the only driver of protein accumulation, the same signal should be observed in ordinary filopodia and in bacteria-induced TMS surrounding bacterial colonies. The β2AR-YFP accumulated in TMS but not in filopodia (Figs. 2F, 4.3 fold and 1.1 fold, respectively). YFP-CD9 and CD4-YFP were also significantly enriched in *N. meningitidis*-induced TMS (Fig. 2F, CD9: 5.65 fold in TMS (D) compared to 1.3 fold in filopodia (F); CD4: 4.3 fold in TMS compared to 1.1 fold in filopodia), ruling out the possibility that a simple change in membrane shape was the main driver of this enrichment. The signal of the FLIPPER-TR®, a fluorescent hydrophobic lipid membrane probe, was also significantly higher in TMS compared to filopodia (2.18 fold and 0.95 fold, respectively), indicative of some plasma membrane accumulation in the focal acquisition plane of TMS. It remained, however, significantly lower than that of YFP-CD9 (Fig. 2F), suggesting that plasma membrane accumulation on its own accounted only in part for the observed local enrichment of membrane proteins. Overall, the data indicate that in the TMS promoted by *N. meningitidis* T4P, the diffusion of the examined integral membrane proteins is decreased, possibly contributing to the accumulation of specific meningococcal receptors.

## *N. meningitidis*-induced initiation of TMS in host cells is independent of signaling

Ezrin accumulation under bacteria is a hallmark of *N. meningitidis*-promoted receptor signaling in host cells and plays an essential role in stabilizing bacterial adhesion and colony growth. We investigated its potential involvement in the formation of the TMS promoted by T4P. Since ezrin activation and its plasma membrane relocation require its Thr567 phosphorylation[51], EA.hy926 cells were treated prior to bacterial infection with NSC668394[52] (from now on, NSC66), an ezrin Thr567 phosphorylation inhibitor, which selectively blocks ezrin accumulation (Fig. 3A, B) and its downstream effects. In particular, NSC66 treatment inhibits sustained bacterial adhesion after 2 h of infection, which depends on the β2AR signaling pathway and on ezrin accumulation under colonies[9,37,39] (Fig. 3C). Then, using CD9-associated fluorescence as a quantitative marker, we investigated the potential contribution of ezrin accumulation to plasma membrane tubulation. To take into account the decrease of bacterial adhesion 2 h after infection, we quantified CD9 labeling intensity below bacterial colonies and normalized each value with the respective intensity of the DAPI staining, which was used as a proxy of the colony size. Interestingly, the inhibition of ezrin phosphorylation (Fig. 3D) did not affect the level of CD9 accumulation below wild-type colonies when normalized with the number of bacteria, indicating that initial *N. meningitidis*-promoted TMS develops independently of ezrin activation and accumulation.

T4P retraction, which depends on the PilT ATPase activity, contributes to the generation of mechanical forces that activate the T4P-induced β2AR signaling, leading to ezrin recruitment in host cells[9]. In the *N. meningitidis* ΔpilT strain retraction is turned off[53] while elongation is still active; consequently, ΔpilT bacteria express long and immobile T4P. CD9 accumulation (Fig. 3D) and the formation of TMS (Fig. 3E, F) persisted, although reduced, in cells infected with ΔpilT mutant bacteria compared to wild type 2C4.3 meningococci. Similar to what was observed with the wild-type meningococci, NSC66 did not further decrease CD9 accumulation around ΔpilT mutant bacteria (Fig. 3D). Moreover, CD9 mobility in TMS, whether promoted by wild type or by ΔpilT bacteria, was comparable in FRAP experiments (Fig. 3G). Based on these results, the cellular process leading to the formation of TMS appeared independent of both T4P-induced receptor signaling and ezrin phosphorylation and activation. Of note, the persistence of some TMS in the absence of any T4P retraction and ezrin accumulation raises the hypothesis that some purely physical, signaling-independent and T4P retraction-independent mechanism might be involved in this phenomenon.

## Tubulation of the plasma membrane along T4P is maintained when biochemical activity is inhibited in host cells

The shape of the apical membrane in endothelial cells is dynamically controlled by the cytoskeleton and by receptor-driven signaling. To impede any biochemical reaction that would interfere with plasma membrane dynamics, we examined the formation of *N. meningitidis*-induced TMS in cells prefixed 15 min with 4% paraformaldehyde (PFA).

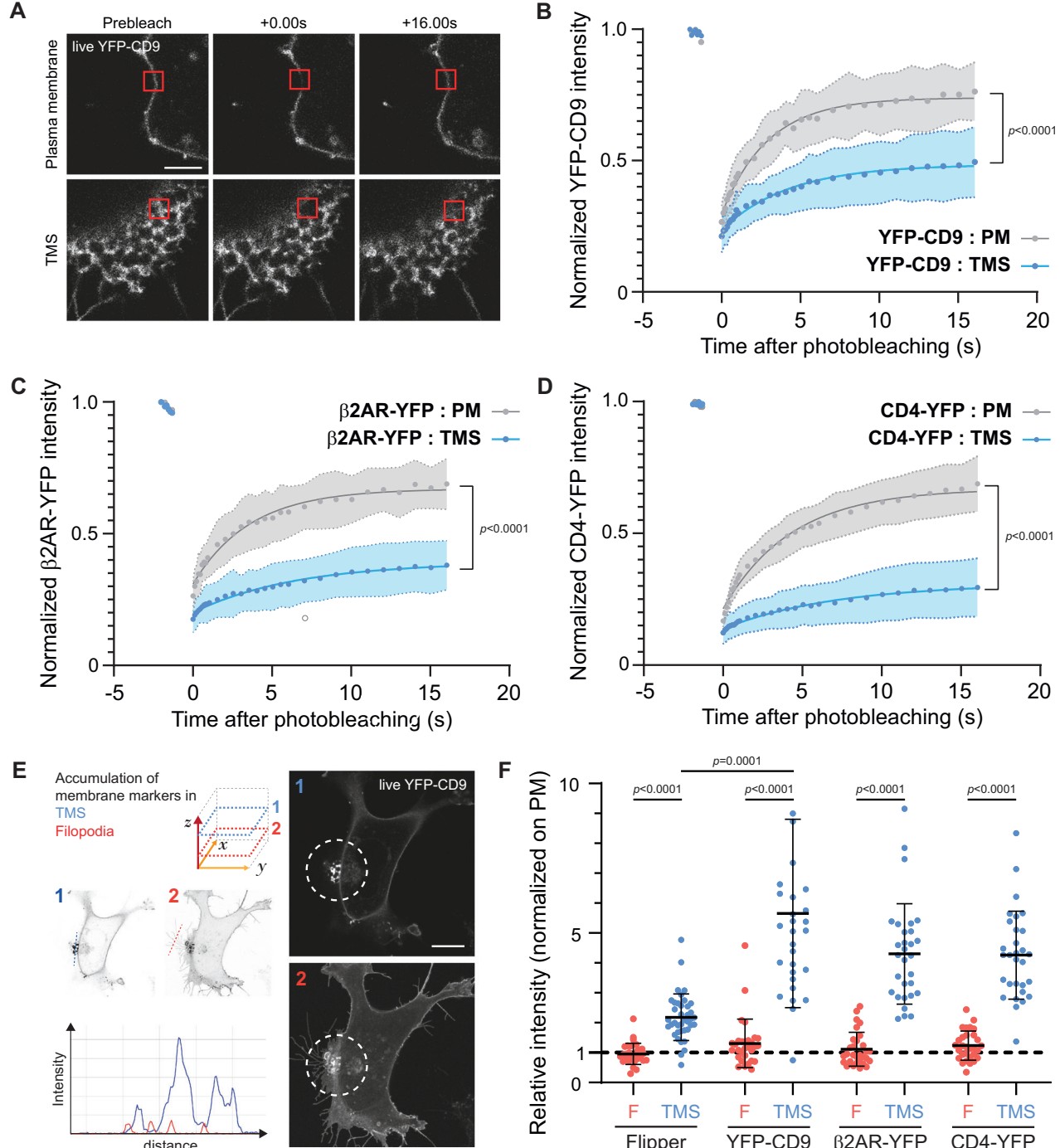

**Fig. 2 | *N. meningitidis* promotes the sequestration of transmembrane proteins in host plasma-membrane protrusions. A–D** FRAP experiments of fluorescent transmembrane proteins expressed in EA.hy926 cells infected with the wild type 2C4.3 strain. **A** Representative FRAP images sequence (one of the thirty cells acquired) of the plasma membrane at the cell periphery (top) and tubular membrane structures (TMS) (bottom) of living EA.hy926 cells expressing YFP-CD9. A 1 μm² region of interest (ROI, red square) was photobleached with high laser power, and fluorescence recovery was monitored within the ROI over time. Scale bars 2 μm. **B** Normalized YFP-CD9, **C** β2AR-YFP and **D** CD4-YFP fluorescence intensity in plasma membranes (PM, grey) and TMS (blue) over time were plotted. Thirty cells from 3 independent experiments were analyzed per condition. Data are presented as means ± SD. Two-sided unpaired *t*-test (at 16 s). **E, F** Quantification of

transmembrane proteins accumulation in *N. meningitidis* induced tubular membrane structures. **E** Example of representative living EA.hy926 cells accumulating YFP-CD9 in tubular membrane structures (one of the thirty cells acquired). Scale bars 10 μm. **F** Maximum intensity values in tubular membrane structures (TMS, blue) and filipodia (**F**, red) were acquired on ImageJ from a trace line and normalized with plasma membranes values and quantified in (**F**). Filopodia and TMS were studied on two different horizontal slices. YFP-CD9 or β2AR-YFP or CD4-YFP were transfected in EA.hy926 cells. The membrane was stained using the Flipper probe that has been added just before infection. Thirty cells from 3 independent experiments were analyzed per condition. Data are presented as means ± SD. Kruskal-Wallis test with Dunn's correction.

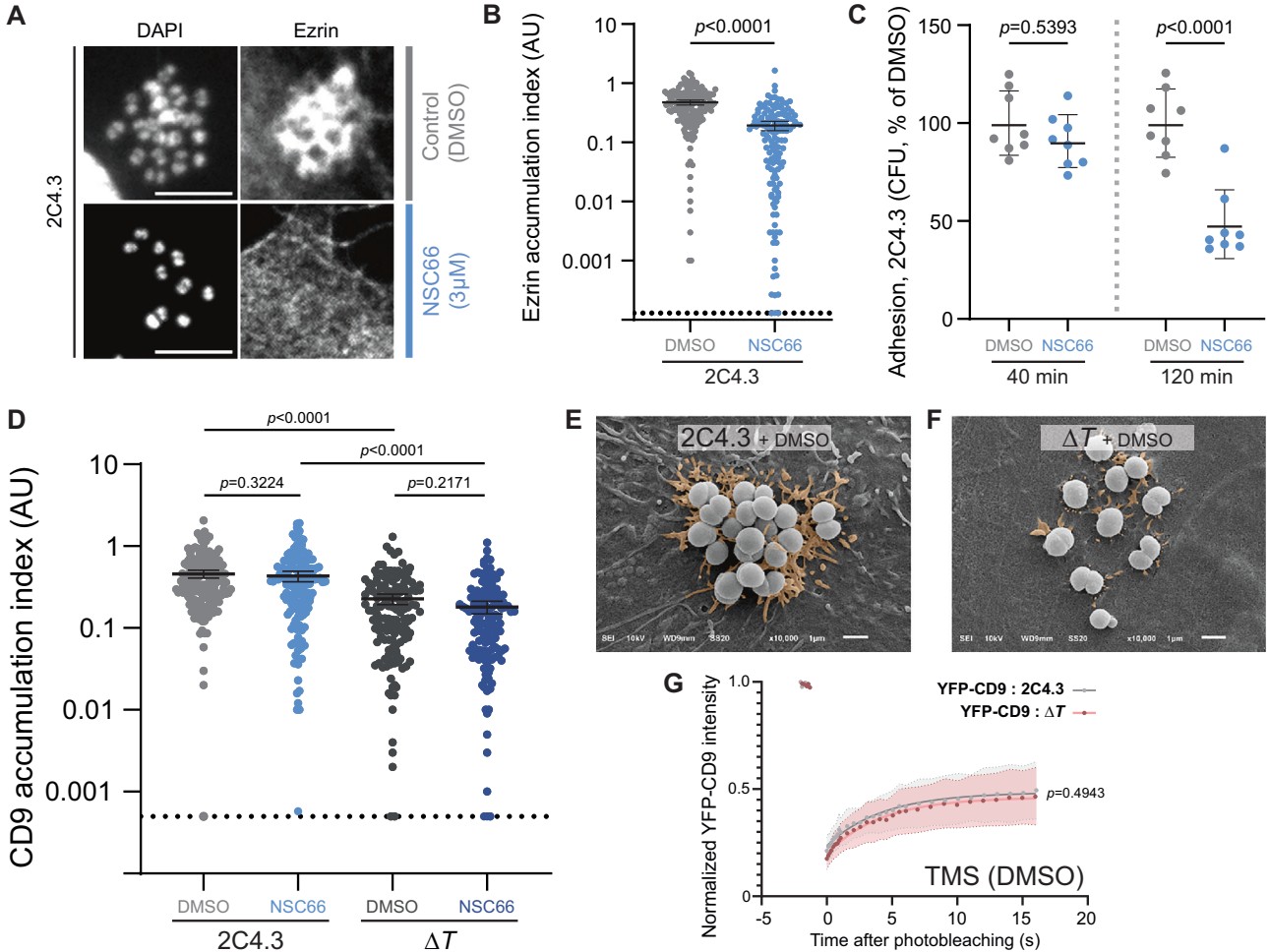

**Fig. 3 | Initiation of tubular membrane structures in host cells is independent of *N. meningitidis* induced-signaling. A–C** Impact of the inhibition of Ezrin Thr567 phosphorylation (3 μM; NSC668394 (NSC66)) in endothelial cells after infection with wild type meningococci (2C4.3). **A** Representative images of endothelial cells immunostained for Ezrin. Z-stack sum projection. Scale bar 5 μm. **B** Quantification of Ezrin accumulation below meningococcal colonies, normalized with DAPI fluorescence (Ezrin accumulation index) in DMSO- or NSC668394-treated endothelial cells. Dashed line represents the detection threshold. Three experiments were pooled. Data are presented as means ± 95% confidence intervals; Mann Whitney test. **C** Adhesion experiment of *N. meningitidis* strain 2C4.3 on DMSO- or NSC668394-treated endothelial cells. CFU mean percentage of the cells treated with DMSO only. Three independent experiments made in duplicate or triplicate

were pooled. Data are presented as means ± SD; One way ANOVA with Bonferroni correction. **D** Quantification of CD9 accumulation below wild type (2C4.3) or its derivative mutant for *pilT* (Δ*T*) meningococcal colonies in DMSO- or NSC668394-treated endothelial cells. Three experiments were pooled. Data are presented as means ± 95% confidence intervals; Kruskal-Wallis test with Dunn's correction. **E, F** Representative SEM images of endothelial cells infected with 2C4.3 or its Δ*T* derivative mutant after treatment with DMSO (three independent experiments). TMS were digitally colored in orange. Scale bar 1 μm. **G** FRAP experiments of YFP-CD9 expressed in EA.hy926 cells infected with 2C4.3 strain or with Δ*T* strain. FRAP was performed in TMS of cells treated with DMSO (grey). Thirty cells from 3 independent experiments were analyzed per condition. Data are presented as means ± SD. Two-sided unpaired *t*-test (at 16 s).

Upon infection of PFA-treated cells with wild type meningococci, some TMS containing endogenous CD9 still developed at the interface with bacteria (Fig. 4A), although much less numerous (Fig. 4D, E) and shorter (Fig. 4C, E) than in living cells. As expected, no ezrin accumulation was observed in prefixed cells (Fig. 4A).

Unexpectedly, upon infection with the Δ*pilT* mutant strain, the accumulation of CD9 around bacterial colonies was much more abundant in PFA-fixed than in live cells, comparable to that observed in living cells infected with the 2C4.3 strain (Fig. 4A, B). CD9 diffusion was similarly impaired by PFA fixation, whatever the infecting strain, both in TMS and plasma membrane filopodia (Fig. 4H, I). Thus, the larger CD9 accumulation in Δ*pilT*-induced TMS of fixed cells (Fig. 4J, about two-fold higher CD9 median intensity in tubular membrane structures, than in filopodia) is likely due to membrane accumulation and not to decreased diffusion. Moreover, in fixed cells infected with Δ*pilT* bacteria, not only the maximal length of TMS was greater than that measured in fixed or living cells infected with the wild type strain (Fig. 4C),

their number was also larger than in living cells, or fixed cells infected with the wild type strain (Fig. 4 compare G with E and F). Importantly, plasma membrane receptors such as CD4 and β2AR, were also accumulated in TMS of fixed cells infected with bacteria (Supplementary Fig. 5). Overall, these findings indicate that the TMS growing along T4P do not require any cell signaling and that they are limited/interrupted by pilus retraction, as indicated by the enhanced maximal length of TMS observed with Δ*pilT* bacteria. Also, in living cells, membrane tubulation along immobile T4P of Δ*pilT* bacteria appears restrained by some fixation sensitive host cell factor(s) (Fig. 4B, C).

To extend our observations in a more representative experimental setup, purified plasma membrane (PM) sheets[54,55] of EA.hy926 cells were prepared from a monolayer of living endothelial cells and then attached to the bottom of Ibidi μ-slides coated with anti-clathrin antibodies. The topology of clathrin-coated zones (flat lattices and coated pits) at the inner side of the plasma membrane facilitates the attachment of PM sheets with the extracellular side oriented upwards.

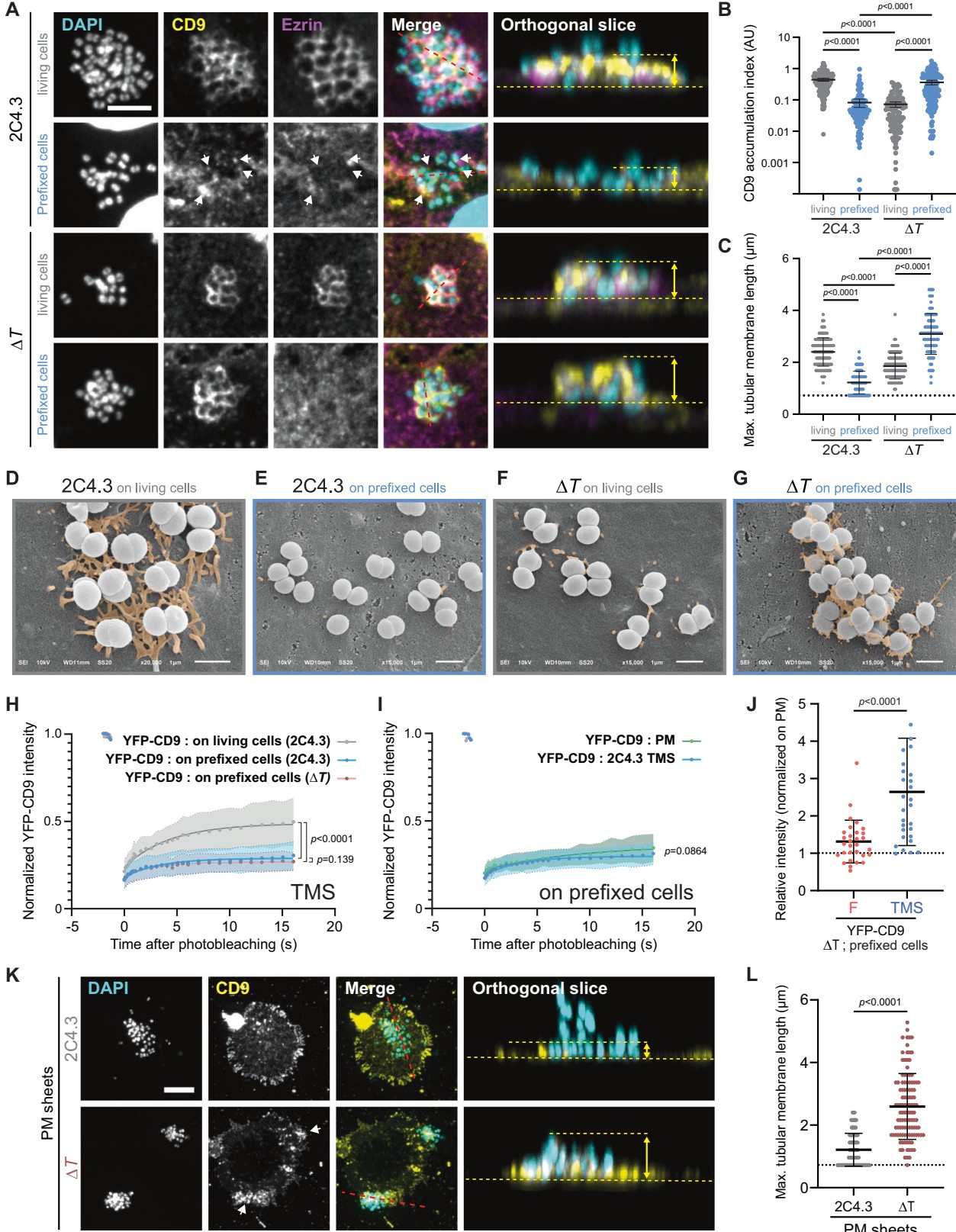

This attachment, which is limited to a few anchoring points, is thought to allow the lipid fraction of the plasma membrane to move freely, while biochemical reactions such as ezrin phosphorylation and actin polymerization are abolished[54,55]. PM sheets were then incubated with wild-type or Δ*pilT* bacteria and the maximal length of TMS was determined (Fig. 4K, L). Consistent with the findings in fixed cells, CD9 accumulation was hardly detectable below wild-type bacteria but much more visible below Δ*pilT* colonies, with the maximal length of TMS induced by Δ*pilT* bacteria being similar to that observed on fixed cells (2.59 μm and 3.09 μm, respectively).

**Fig. 4 | Plasma membrane tubulation depends only on the plasma membrane dynamics. A–G** Infection of living (grey) or prefixed with 4% PFA (blue) EA.hy926 endothelial cells with the 2C4.3 strain or its derivative deletion mutant for *pilT* (Δ*T*). **A** Representative images of CD9 and ezrin immunostaining at the site meningococcal adhesion. CD9 (yellow) remains present in fixed cells in the absence of ezrin (magenta). DAPI was used to reveal bacterial DNA (cyan). On the left part: Z-stack sum projection. On the right part: Orthogonal slices. White arrows indicate puncta of CD9 accumulation. The red dashed lines indicate the position of the orthogonal slices. The yellow arrows and yellow dashed lines were added to represent the height of tubular membrane structures. Scale bar 5 μm. **B** Immunofluorescence quantification of CD9 accumulation below the meningococcal colonies, normalized with DAPI fluorescence of the corresponding colonies. Three experiments were pooled and analyzed with Kruskal-Wallis test with Dunn's correction. Data are means ± 95% confidence intervals. **C** Maximum height of TMS were estimated from CD9 immunofluorescence imaging. The orthogonal slices in A were only added to help visualizing the membrane structures. At least 90 colonies were studied from three independent experiments. One-way ANOVA with Bonferroni's correction. Data are means ± SD. **D–G** Representative SEM images of infected endothelial cells. TMS were digitally colorized in orange (three independent experiments). Scale bar 1 μm. **H, I** FRAP Experiments of YFP-CD9 expressed in EA.hy926 cells. Thirty cells from 3 independent experiments were analyzed per condition. Data are presented as means ± SD. Two-sided unpaired *t*-test (at 16 s). **H** FRAP experiments on TMS in living cells infected with the 2C4.3 strain (grey), prefixed cells infected with the 2C4.3 strain (blue) or its derivative deletion mutant for *pilT* (Δ*T*, dark red). **I** FRAP experiments on plasma membrane (PM, green) or tubular membrane structures (TMS, blue) of prefixed cells infected with 2C4.3. **J** Quantification of YFP-CD9 protein accumulation in *N. meningitidis* Δ*T* induced membrane structures, as in Fig. 2F. Maximum intensity values in tubular membrane structures (TMS, blue) and filipodia (F, red) were acquired on ImageJ from a trace line and normalized with plasma membranes values and quantified in prefixed EA.hy926 cells. Thirty cells from 3 independent experiments were analyzed per condition. Data are presented as means ± SD. Mann Whitney test. **K, L** TMS in PM sheets of EA.hy926 endothelial cells infected with the 2C4.3 strain (grey) or its derivative deletion mutant for *pilT* (Δ*T*, dark red). **K** Representative images of CD9 immunostaining (yellow) at the site meningococcal adhesion. DAPI was used to reveal cell and bacterial DNA (cyan). Z-stack sum projection. White arrows indicate CD9 accumulation. Scale bar 5 μm. **L** Quantification of YFP-CD9 protein accumulation in *N. meningitidis* induced tubular membrane structures, as above. At least 85 colonies were studied from three independent experiments. Data are presented as means ± SD. Two-sided unpaired *t* test with Welch's correction.

## Actin polymerization inhibits tubulation of the plasma membrane induced by *N. meningitidis* T4P

From the data above, it appeared that some host cell factors limit the formation of TMS promoted by immobile T4P. To address the possibility that this might be due to the reaction of the cortical cytoskeleton to a local increase in membrane tension, we examined whether the inhibition of membrane tubulation along T4P would require ATP as energy source. Endothelial cells were incubated with 10 mM 2-deoxy-d-glucose (2-DG) to deplete cellular ATP (Supplementary Fig. 6), before meningococcal infection. The length of CD9-labeled TMS was then estimated using microscopy orthogonal sections (Fig. 5A,B). ATP depletion, to levels sufficient for inhibiting ezrin accumulation (Fig. 5A), significantly reduced the length of protrusions in cells infected with the wild-type 2C4.3 strain (Fig. 5B), confirming the requirement of host cell signaling in the formation of mature protrusions in and around the colony. However, reminiscent of what was observed on PM sheets or upon PFA fixation of endothelial cells (see Fig. 4), ATP depletion markedly enhanced the length of TMS induced by the Δ*pilT* strain infection (Fig. 5A, B). Actin polymerization, typically observed under bacterial colonies in infected endothelial cells[39,44], requires ATP. Pre-treatment of endothelial cells with the actin polymerization inhibitors Cytochalasin D (CytD) and Latrunculin B (LatB) enhanced the maximum length of TMS promoted by Δ*pilT* bacteria to the same level of low ATP conditions (Fig. 5B, C). Altogether, our findings indicate that the signaling-independent growth of TMS is actually counteracted by actin cytoskeleton remodeling in host cells.

## *N. meningitidis* can induce TMS in cells incompetent for meningococcal-promoted signaling

To further validate the hypothesis that TMS induced by T4P involves a purely physical mechanism, additional studies were conducted in cell types, that are not competent for meningococcal infection, meningococcal-induced signaling, or ezrin recruitment. Kidney epithelial HEK cells cannot be infected by wild-type *N. meningitidis* because bacteria cannot adhere to their surface; in case of surrogate adhesion, bacteria do not induce any signaling in the absence of exogenous β2AR and β-arrestins[7]. Surrogate adhesion of a *N. meningitidis* *SiaD*⁻ *Opa*⁺ strain can be induced in HEK cells by expressing hCEACAM1, the adhesion receptor of opacity proteins (Opa)[44,50]. In hCEACAM-expressing HEK cells, CD9 - used as a marker of T4P-induced TMS - accumulated in and around adherent *SiaD*⁻ *Opa*⁺ Δ*pilT* colonies (Fig. 6A, left panels). Control non-piliated Δ*pilE* bacteria failed to induce any CD9 accumulation (Fig. 6A, middle panels), consistent with the absolute requirement of T4P in this process. *SiaD*⁻ *Opa*⁺ Δ*pilT*

colonies also induced the accumulation of CD9 in canine MDCK hCEACAM cells, in which no meningococcal-promoted signaling can be observed (Fig. 6A, right panels). As shown by FRAP experiments in endothelial cells, YFP-CD9 mobility was decreased in infected HEK cell membrane structures surrounding bacteria, compared to control distant plasma membrane areas (Fig. 6B, compare with Fig. 2B). Furthermore, CD9 accumulation around bacteria was maintained in PFA-fixed HEK cells (Fig. 6C), recapitulating what was observed in endothelial cells. These findings support the hypothesis that adhesive *N. meningitidis* T4P generate a purely physical process, independent of β2AR signaling, driving plasma membrane tubulation.

## PilC1-dependent adhesion is the initiating event of membrane tubulation

The data accumulated so far did not determine what component of T4P could be the initial driver of the tubulation process in host cells. The PilV pilin and the tip-located PilC1 adhesin are both required for adhesion, CD9 recruitment and membrane tubulation (Fig. 1D, E). PilV interacts with CD147 and β2AR receptors and activates T4P retraction-induced signaling in host cells, suggesting that PilV is mainly involved in meningococcal adhesion after receptors accumulation in TMS. Although also involved in adhesion, PilC1 has no known signaling properties. PilC1 thus appears as a strong candidate for the initiation of membrane tubulation in host cells. The genetic deletion of PilC1 in wild type or Δ*pilT* strains (Δ*C1* and Δ*TC1* mutants, respectively) significantly and specifically (rescue experiments, "+C1") inhibited meningococcal adhesion in both living and PFA-fixed cells (Fig. 7A). The preserved expression of PilV in T4P of Δ*TC1* bacteria was therefore not sufficient for maintaining normal adhesion (Fig. 7A, B) and could not rescue membrane tubulation in *SiaD*⁻ *Opa*⁺ Δ*C1* bacteria (Fig. 1D, E). Also, the significant enhanced adhesion specifically observed in fixed cells infected by the Δ*pilT* strain, was not observed in case of concurrent absence of PilC1 (Fig. 7A), despite the advantage provided by PilT suppression in terms of prolonged T4P contact with host cells plasma membrane and T4P length (see Fig. 4). All together, these data demonstrate a predominant role of PilC1 as the inducer of TMS in host cells at the initial steps of adhesion.

## Discussion

We uncovered the mechanism adopted by *N. meningitidis* to facilitate the encounter with cognate host-cells receptors despite the initial low multiplicity of infection usually observed with this pathogen. Bacterial T4P interact through the pilus tip adhesin PilC1 with host cells, inducing an early signaling-independent host plasma membrane tubulation

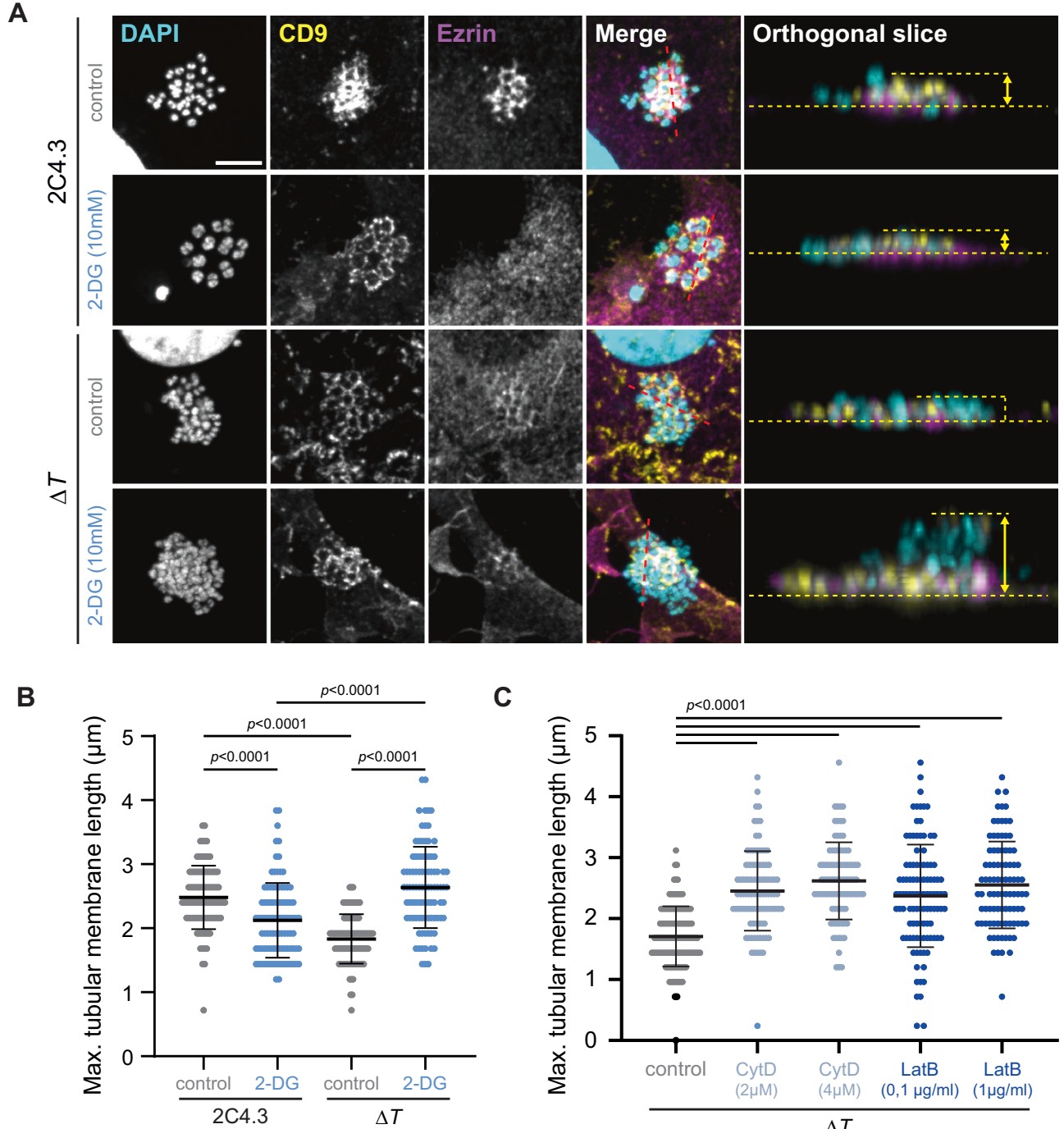

**Fig. 5 | Actin depolymerization frees plasma membrane tubulation upon infection with ΔpilT strain. A–C** Infection of EA.hy926 endothelial cells with the 2C4.3 strain or its derivative deletion mutant for *pilT* (Δ*T*), with 2-DG (10 mM, blue) or without (control, grey). **A** Representative images of CD9 and ezrin immunostaining at the site meningococcal adhesion. On the left part: Z-stack sum projection. On the right part: Orthogonal slices. The red dashed lines indicate the position of the orthogonal slices. The yellow arrows and yellow dashed lines were added to represent the height of plasma membrane protrusions. Scale bar 5 μm. Maximum height of TMS were estimated from CD9 immunofluorescence imaging. Cells were infected after treatment with 2-DG (**B**) or cytochalasin D (CytD, light blue) and Latruncumin B (LatB, dark blue) (**C**) or without (control, grey). The orthogonal slices in A were only added to help visualize the height of plasma membrane protrusions. At least 90 colonies were studied from three independent experiments. Kruskal-Wallis test with Dunn's correction. Data are means ± SD. Brown-Forsythe and Welch ANOVA with Games-Howell's correction.

(Fig. 8, step 1). These TMS extend the interaction surface between meningococcal ligands, such as PilV molecules with potential receptors. The plasma membrane-associated receptors found in these tubular structures exhibit reduced mobility, compared to areas of the plasma membrane distant from infecting bacteria. The increased local concentration of potential adhesion and signaling receptors in the vicinity of bacterial ligands eventually enable PilV to bind to CD147 / β2AR oligomers during the short time frame (few seconds) of pili immobility, before PilT-dependent retraction. PilV interaction with CD147 facilitates additional PilV molecules distributed along the pilus to bind to glycan chains of β2AR signaling receptors (Fig. 8, step 2). The mechanical forces resulting from both T4P retraction and blood flow,

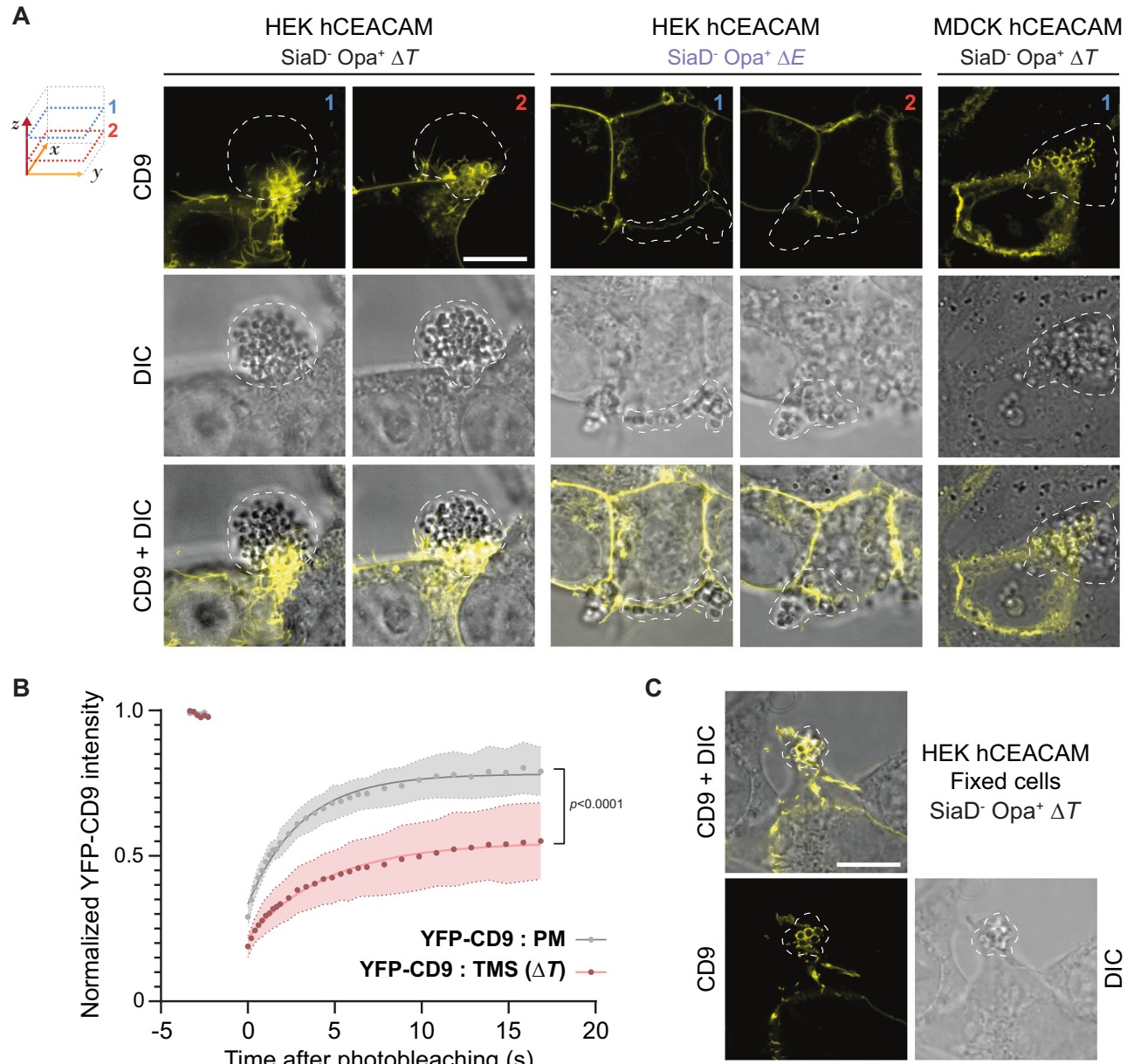

**Fig. 6 | *N. meningitidis*-induced tubular membrane structures can be recapitulate in cells incompetent for meningococcal induced signaling.**
**A** Representative confocal and DIC images of HEK cells or MDCK cells expressing YFP-CD9 (yellow) infected with the non piliated 2C4.3 *SiaD⁻ Opa⁺ ΔE* strain or the non-retracting 2C4.3 *SiaD⁻ Opa⁺ ΔT* strain (two independent experiments). Confocal images were acquired on different z positions: 1 (on the top of cells, blue) and 2 (on the bottom of cells, red). Bacterial colonies are indicated by white dashed shapes. Scale bar: 10 μm. **B** FRAP Experiments of YFP-CD9 expressed in EA.hy926 cells on plasma membrane (PM, grey) or tubular membrane structures (TMS, dark red) in living cells infected with the *pilT* mutant (Δ*T*). Thirty cells from 3 independent experiments were analyzed per condition. Data are presented as means ± SD. Two-sided unpaired *t*-test (at 16 s). **C** Representative confocal and DIC images of prefixed HEK cells expressing YFP-CD9 (yellow) infected with the 2C4.3 *SiaD⁻ Opa⁺ ΔT* strain (two independent experiments). One representative z position has been selected. Scale bar: 10 μm.

applied on the β2AR via PilV, trigger signaling in host cells. The subsequent signaling- and ATP-dependent enrichment of TMS with ezrin and actin polymers allows their enlargement and reinforcement. The cellular protrusions resulting from this process stabilize meningococci at the cell surface, permitting the interaction of additional T4P ligands with additional receptors, and then the progressive growth of a bacterial colony despite the forces exerted by the blood flow (Fig. 8, step 3).

The ability of the plasma membrane to form early tubular structures along meningococcal T4P was previously observed, but its functional relevance in the context of *N. meningitidis* pathophysiology had not been addressed so far. The formation of these tubular structures was described as a physical process of one-dimensional "wetting" occurring when membranes interact with nanofibers[36]. The physical nature of this phenomenon is supported here by the demonstration that it is independent of meningococcus-induced signaling, it does not require ATP as energy source and can be reproduced in fixed cells or with purified plasma membrane sheets. Our data also point out the central role of the PilC1 pilus-tip adhesin docking to the plasma membrane, in the initiation of the one-dimensional wetting. In line with our hypothesis, the host cell "factor" interacting with PilC1 should be extremely abundant at the cell surface to facilitate rapid interaction with the tip-associated adhesin. This factor remains unknown to date. Although CD46 has been proposed as an adhesion

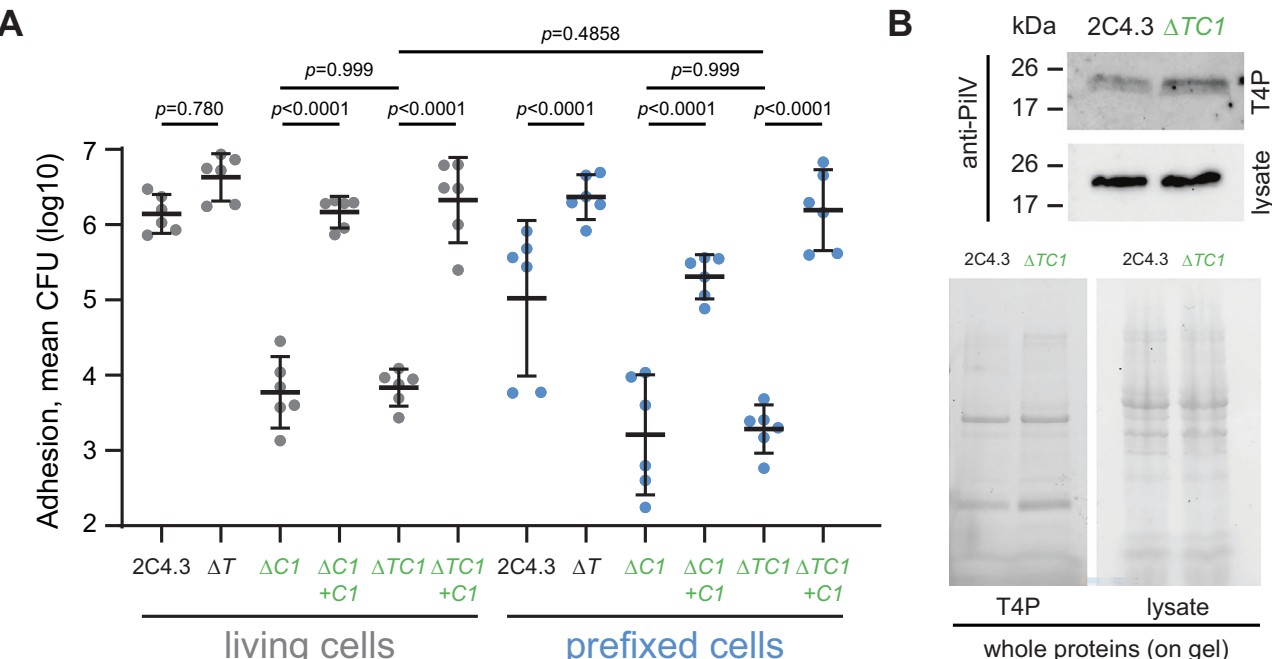

**Fig. 7 | PilC1-dependent adhesion is not rescued by PilT mutation, despite PilV expression. A** Adhesion experiment of *N. meningitidis* strain 2C4.3 or its derivate mutant for *pilT* (Δ*T*),*pilC1* (Δ*C1*) or both *pilT* and *pilC1* (Δ*TC1*) or the complemented mutant for *pilC1* (+C1) on living (grey) or prefixed (blue) EA.hy926 endothelial cells. Mean CFU. Three independent experiments performed in duplicate were quantified per condition. Data are presented as Geometric means CFU ± 95% confidence interval; One way ANOVA with Bonferroni's correction. **B** Expression of PilV in pili preparation (T4P) and whole bacterial lysate (lysate) of the wild type strain 2C4.3 and its derivate mutant for *pilT* and *pilC1* (Δ*TC1*). On the top, western blotting using anti-PilV antibodies. On the bottom, total proteins were detected in the gel by Stain-Free technology (Biorad).

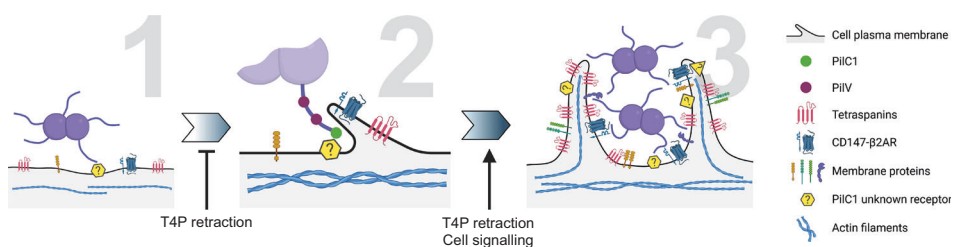

**Fig. 8 | Schematic representation of our proposed model of *N. meningitidis*-cell interaction, integrating adhesion, tubular membrane structures and induced signaling.** From left to right: (i) Meningococci interact with endothelial cells via the PilC1 adhesin. (ii) After interaction with an unknown PilC1 ligand, the cell membrane expands along T4P. This tubulation of the membrane is only possible during the few seconds of T4P immobility. This promotes the accumulation and clustering of receptors, allowing the T4P adhesins to interact with the signaling receptors CD147 and β2AR. Early tubular membrane structures can be revealed in fixed cells or using a pilT deletion mutant. (iii) Finally, T4P pulls on the plasma membrane, activating signaling via the CD147 and β2AR receptors, leading to the accumulation of ezrin and actin and the stabilization of large tubular membrane structures, which are the site of further accumulation of integral membrane proteins. This triggers an amplification loop leading to an increase in the membrane surface area to which bacteria can adhere and be protected from the shear stress of blood flow. Created in BioRender. Laurent, A. (2025) https://BioRender.com/75yfi0h.

receptor, its role in meningococcal infection is still debated, and there is no evidence of any interaction with PilC1. Among other candidates, PilC1 might interact with glycan(s) present at the plasma membrane; this property was reported for other tip-associated adhesins, such as PilB from *S. sanguinis* or FimH the type I pilus adhesin from *E. coli*.

The early tubular structures, promoted by T4P, display several properties that differentiate them from common plasma membrane filopodia. They accumulate multiple integral membrane proteins, including the tetraspanins CD9, CD81 and CD151, G protein-coupled and chemokine scavenger receptors, and the single transmembrane domain proteins CD4, CD44, and CD147. Using CD9 as a marker of these tubular structures, we could determine that its local enrichment around bacteria might not only depend on membrane accumulation - the tubular formation itself - but also on its markedly decreased mobility. Indeed, once in the tubular structure, membrane proteins

can move along the walls of the tube, with a lower probability of exiting from its basis, keeping them trapped. Our study did not address the lipid composition of TMS. Previous work suggested the accumulation of cholesterol[40] and sphingolipids[56,57], which could alter the viscosity and the tension of the membrane at the TMS. Such accumulation might involve active lysosomal exocytosis. Our results (see Fig. 2F) also suggest an active receptor accumulation phenomenon in living cells that has yet to be determined. Moreover, it is not clear how other integral membrane proteins, such as the transferrin receptor or Toll like receptors, are excluded from these structures.

Several factors regulate the size and the lifetime of these T4P-promoted tubular structures in living cells. As mentioned above, PilT-dependent pilus retraction limits their duration to a few seconds[33], unless signaling events promoted in host cells stabilize and turn them into larger cellular protrusions. In this context, ezrin recruitment and

phosphorylation play an essential role for adhesion and protrusion stabilization 2 h after infection. The role of cortical actin polymerization is more complex, since it has a variable effect depending on the duration of T4P interaction with the host cell plasma membrane. On one hand, its inhibition by ATP depletion in cells infected by the wild-type strain (which retracts its T4P) reduces the maximal size of tubular structures. On the other hand, in cells infected with the ΔpilT strain, the inhibition of actin polymerization, either indirectly through ATP depletion or directly using cytochalasin D or latrunculin B, markedly enhances the length of tubular structures.

In addition to *Neisseria* family bacteria, many other species express fimbriae or pili for adhesion and infection of host cells. Interestingly, pili-tip adhesin proteins, such as *Pseudomonas* PilY1, uropathogenic *E. coli* FimH, and *Streptococcus pneumoniae* RrgA are similarly exposed and directly engaged in the interaction with host cells by targeting specific receptors or extracellular matrix (ECM) components. Since the pili-dependent formation of TMS appears to be a physical process of one-dimensional wetting resulting from the interaction of membranes with nanofibers (i.e. pili), it is plausible that the mechanism of cognate receptor recruitment in these tubular structures for *N. meningitidis* extends to other bacterial species. Mammalian cells can also produce nanofibers, including actin-based structures (filipodia, microvilli) and ECM components (collagen, fibronectin), which play a role in cell signaling and adhesion. It is not excluded that these structures might induce similar membrane tubular processes in adjacent/interacting cells.

# Methods

## Cell lines and culture conditions

**Cell lines and media.** The EA.hy926 cell line (ATCC #CRL-2922) was obtained from the American Type Culture Collection (ATCC). EA.hy926 cells were grown in Dulbecco's Modified Eagle Medium (DMEM; Gibco) supplemented with 10% decomplemented fetal calf serum (FCS; Gibco) and 1% penicillin–streptomycin. Human Cerebral Microvascular Endothelial Cells hCMEC/D3s[58] are fully differentiated brain endothelial cell derived from human brain capillaries. hCMEC/D3 were grown onto cultrex rat collagen type I-coated dishes (Bio-techne) in Endothelial Cell Basal Medium-2 (Lonza) supplemented with 5% of FCS, 1.4 μM hydrocortisone (Lonza), 5 μg/mL ascorbic acid (Lonza), 1 ng/mL b-FGF (Lonza), at 37 °C in 5% CO2. Primary Human Dermal Blood (Microvascular) Endothelial Cells (HDBEC) (Promocell) are isolated from the dermis of juvenile foreskin and adult skin (different locations). HDBEC were grown onto cultrex rat collagen type I-coated dishes (Bio-techne) in their specific endothelial cell growth medium (PromoCell). HEK-293 (ATCC CRL-1573) cells were grown in flasks coated with 0.01% poly-l-lysine (Sigma) in supplemented with 10% FCS and 1% penicillin–streptomycin. Cells were grown at 37 °C in a humidified incubator under 5% CO2.

**Transfection and siRNA inhibition.** EA.hy926 cells were transfected with 1 μg of plasmids using Lonza 4D Nucleofector® System (Lonza) according to manufacturer's instructions and $5.10^5$ transfected cells were seeded into μ-Slide 4 Well (Ibidi) 24 h prior to infection. For HEK cell, $1.5 \times 10^5$ cells were seeded in μ-Slide 4 Well and were transfected 24 h prior to infection with 200 ng of plasmid encoding hCEACAM and YFP-CD9, using EcoTransfect™ (Oz Biosciences) according to manufacturer's instructions. For relative intensity and accumulation quantification experiments, infections were performed in Opti-MEM (Gibco). The siRNA inhibitions of CD9, CD81 and CD151 were obtained from siRNA pools (ON-TARGETplus Human CD9 (928) siRNA - SMARTpool (Horizon); ON-TARGETplus Human CD81 (975) siRNA - SMARTpool (Horizon); MISSION Pre-designed siRNA −2 OD, Human CD151, SASI_Hs01_00188401 (Sigma)) and controlled with MISSION® siRNA Universal Negative Control #1 (Sigma). The siRNAs were transfected with Lipofectamine RNAiMAX (Thermofisher) on non-confluent

cells. For CD9 siRNA, the transfection was repeated a second time after 3 days. Experiments were performed 3 days after the last transfection. Because tetraspanins are accumulated below meningococcal colonies, the whole amount of protein in cells did not reflect the true level of protein accumulation. Therefore, the inhibitions of tetraspanins expression were controlled by immunofluorescence quantification under bacterial colonies (see the image analysis section below).

## Production of the CRISPR/Cas9 knockout cell lines

**Cell line mutation.** The triple knockout (KO) EA.hy926 cell line was obtained by serial knockout of CD9, then CD81 and then CD151. CD9 and CD81 KO cell lines were obtained using the lentiCRISPR v2 plasmid (gifts from Dr Eric Rubinstein; Addgene plasmid #52961). The CD151 KO cell lines were obtained using the Sigma-Aldrich CRIPRD HSPD0000006331 and HSPD0000006332 plasmids. For each of the three separate tetraspanins we produced two lentiviruses with different guiding sequence. After obtaining each KO cell lines, tetraspanin depletion, adhesion of bacteria and induced signaling were assessed. One cell line was kept for further transduction. Guiding sequences are as follow: CD9sg95 TTGGACTATGGCTCCGATTC & CD9sg314 ATTCGCCATTGAAA-TAGCTG; CD81sg166 ACACCTTCTATGTAGGTGAG & CD81sg253 AGGAATCCCAGTGCCTGCTG; CD151 n°6331 CTGGTAGTAGGCG-TAGGCG & CD151 n°6332 CCAAGCGCTACCACCAGCC. EA.hy926 cells were transduced with lentivirus (MOI 1:1000). 48 h after transduction, cells were treated with 1 μg/ml puromycin to kill non-transduced cells. FACS sorting of tetraspanin-depleted cells was performed 2 to 3 times over the scope of 1 or 2 months to ensure the purity of each CRISPR KO cell lines. Note that we did not select a clonal population but the whole population of the CD9, CD81 and CD151 depleted cells.

**Cytometry.** For surface protein expression analysis, cells were washed twice in Phosphate-Buffered-Saline (PBS), trypsinized, fixed in 4% paraformaldehyde (PFA) for 15 minutes, washed twice in PBS and kept at 4 °C until staining. For cell sorting, cells were washed twice in PBS, trypsinized, and immediately washed twice in PBS before staining. In both experiments, cells were stained for 30 minutes with the appropriate primary antibody at 4 °C in PBS/Bovine Serum Albumin (BSA) 0.1% and washed three times in PBS-BSA before Alexa Fluor-coupled secondary antibody staining for 30 minutes. Cells were then washed twice in PBS-BSA before FACS analysis. Negative controls were assessed with secondary antibody-only staining on the same cells. Data were acquired using a BD LSR Fortessa instrument (BD Biosciences) for fixed samples, and with BD FACSAria II for live cell sorting. Data was analyzed using the FlowJo Software. A minimum of 20 000 cells were acquired for each experiment.

## Treatments of cells

**Actin polymerization and ezrin inhibition drugs.** Treatments with drugs were performed in DMEM + 10% FBS for NSC668394 (Sigma-Aldrich 341216; 30 μM 2 h prior to infection, and 3 μM throughout infection), wheat germ agglutinin (WGA; Vector Laboratories, Burlingame, CA, USA; 20 μM, 1 h prior to infection and maintained throughout infection), Cytochalasin D (Sigma C8273; 2-4 μM, 2 h prior to infection and maintained throughout infection), Latrunculin B (Abcam ab141409; 0,1–1 μg/ml, 2 h prior to infection and maintained throughout). Controls were assessed with DMSO when relevant.

**ATP depletion.** For ATP depletion, treatment was performed in DMEM without glucose (Invitrogen) + 1 mM lactate (as carbon source for bacteria) and 10 mM 2-deoxy-d-glucose (2-DG, Sigma-Aldrich). Control cells were incubated in DMEM without glucose, supplemented with 10% FBS. Treatment was performed overnight prior to infection and maintained through the course of the experiment. ATP depletion efficiency was assessed using the commercially available kit Adenosine 5′-triphosphate Bioluminescent Assay (Sigma-Aldrich).

**Chemical pre-fixation assay.** Cells were chemically fixed on the day of the infection with 4% paraformaldehyde (PFA, Thermofisher 28908) for 15 minutes and quenched with 50 mM NH₄Cl for 5 minutes. Cells were then washed several times in PBS and DMEM + 10% FBS prior to infection.

**Bacterial strains and infection**

*N. meningitidis* strains used in this study are derived from serogroup C meningococcal strain 8013, designated as 2C4.3[59]. 2C4.3 is a piliated encapsulated Opa− Opc− variant. Mutant strains ΔpilC1[31], ΔpilE[8], ΔpilT[4], ΔpilV[38], ΔsiaD[44] as well as double mutants were designed in our laboratory. Complemented mutant expressing *pilC1* under the control of the *tet*-promoter were obtained after transformation of the pNM99-pilC1 integrative plasmid in the relevant strains. The pNM99 plasmid was engineered in our laboratory[60] and the *pilC1* sequence was cloned from the 2C4.3 genome and inserted into the pNM99*tet* plasmid using the NEBuilder® HiFi DNA Assembly Cloning Kit (New England Biolabs) and the following primers to amplify the plasmid and *pilC1* pNM99*tet*: pNM99_Fw: GCGGTGGCGGCCGCTC; pNM99_Rv: GTGGAGCTCCAATT GGCCC; PilC1_ Fw: AGGGCCAATTGGAGCTCCACATGAATAAAACTTT AAAAAGGCAGG; PilC1_ Rv: TCTAGAGCGGCCGCCACCGCTCAGAA-GAAGACTTCACGCCAGCTG. *N. meningitidis* were grown at 37 °C in 5% CO₂ on gonococcal base (GCB) agar (Difco) plates containing 12 μM FeSO4 and Kellogg's supplements[61] or in Dulbecco's Modified Eagle Medium (DMEM; Gibco) supplemented with 10% decomplemented fetal calf serum (FCS; Gibco). For antibiotic selection of *N. meningitidis* strains, kanamycin was used at a concentration of 100 μg/ml, chloramphenicol at 6 μg/ml and erythromycin at 2 μg/ml. Before cell infection, bacteria were first sub-cultured to OD600 = 0.1 in pre-warmed cell culture medium and incubated for 2 h at 37 °C with agitation and 5% CO₂.

**Imaging assay.** Cells were infected with bacteria at a multiplicity of infection of 30 bacteria per cell for 30 min and washed with media to remove non-adherent bacteria. Infection then proceeded for 2 h and was concluded with two PBS washes and 4% PFA fixation for 15 minutes.

**Bacterial adhesion assay.** At least two days before infection, cells were seeded on 2 cm² wells. The day of infection, confluent adherent cells were infected with an estimated number of 10⁷ bacteria each for 40 min. The exact number of colony forming units (CFU) in the inoculum was determined by serial dilution and counting of CFU. After 40 min, the cells were washed six times with PBS to remove non-adherent bacteria and adherent bacteria were detached in 500 μL DMEM with 10% FBS by mechanical scratching. To determine the adhesion frequency, adherent bacteria were diluted and spread on agar plates and the CFU were counted the next day. Adhesion frequency was obtained by dividing the adherent bacteria CFU count with the inoculum CFU count. For longer adhesion assay, infection was allowed to proceed for 2 h after the first washes at the 40-minutes mark and was concluded similarly to the early adhesion assay.

**Imaging**

**Immunofluorescence microscopy.** All incubations were performed at room temperature. Cells were grown and infected on glass coverslips. They were fixed in 4% PFA for 15 minutes, quenched with ammonium chloride 50 mM solution for 5 min and rinsed with PBS with ions. Immunostaining steps were performed with 0.1% BSA blocking and 0.1% Saponin permeabilization at all times. Coverslips were incubated in BSA and Saponin buffer for 20 minutes before staining. Proteins of interest were stained with primary antibodies for 45 min to 1 h. After three washes in PBS, the coverslips were incubated with Alexa Fluor-conjugated secondary antibodies for 45 min. Nuclear DNA and actin were stained with 4',6'-diamidino-2-phenylindole (DAPI)

at 1 μg/ml and Alexa Fluor-conjugated phalloidin (Thermofisher), respectively. After three PBS washes, the coverslips were rinsed in water and mounted in Mowiol for observation. Immunofluorescence imaging was performed either on a Zeiss Spinning Disk microscope (63X, NA 1.4), a Zeiss Apotome fluorescence microscope or a SP8 laser scanning confocal microscope (Leica Microsystems) using a oil immersion objective. When mentioned, deconvolution was performed using Huygens software.

**Scanning Electron Microscopy.** All incubations were performed at room temperature except when otherwise indicated. Cells were grown and infected on glass coverslips. They were fixed in 0.1 M sodium cacodylate (Euromedex) buffer, pH 7.4 containing 2.5% glutaraldehyde (GA, Euromedex) and 1% PFA for 60 min, washed in cacodylate buffer (2 × 10 min), and then fixed in 1% OsO4 diluted in cacodylate buffer for 45 min at 4 °C. After washing in cacodylate buffer (2×10 min), samples were dehydrated in an ascending series of ethanol (30%, 50%, 70%, 95%, 100%, 100%, 100% − 10 min each), followed by Hexamethyldisilazane (HMDS, Sigma-Aldrich)/ethanol (1/1:v/v) for 10 min and HMDS for 10 min. After overnight air drying, each coverslip was placed on a double-sided sticky tape on the top of an aluminum stub and sputter-coated with Au/Pd. Images were acquired using a Jeol LV6510 (Jeol, Croissy-sur-Seine, France).

**Correlative Light Electron Microscopy (CLEM).** Cells were grown and infected in 35 mm μ–Dishes with a removable gridded glass coverslip bottom (Ibidi 81158). They were first processed for fluorescence imaging as follows. After fixation in 4% PFA + 0.2% GA, they were immunostained similarly to classic immunofluorescence microscopy. They were then imaged in a SP8 laser scanning confocal microscope (Leica Microsystems) using a X63/1.40 oil immersion objective and 405/410-450, 488/495-530, and 552/560-600 excitation/emission wavelengths (in nm) for the detection of DAPI, AF488, and AF555, respectively. A mosaic image of the whole grid, including the lettered and numbered squares, was acquired in the DAPI channel and used as a map to locate the cells that were then observed at high resolution. For each selected region of interest (ROI), a z-stack with a step of 0.3 μm was acquired and deconvoluted using Huyghens software. Following confocal imaging, cells were fixed and processed for SEM as indicated above. Each ROI previously acquired in the confocal microscope was imaged in the SEM. For correlative image registration between electron microscopy (EM) and fluorescence the Icy (v2.5.2.0)[62] EC-CLEM plugin[63] has been used with the DAPI signal as the referent channel for the registration.

**FRAP.** Live experiment data were acquired with a TCS SP8 X confocal microscope (Leica) equipped with a ×100, 1.40 oil HC PL APO objective and a heated stage maintained at 37 °C. For FRAP analysis, series of 40 fluorescence measures were captured from 30 images for all experiments from three independent experiments, with time intervals between scans ranging from 100 to 1000 ms. To perform FRAP, 1 μm²-ROI were bleached with 10 WLL2 laser pulse iterations at 100% power. Image analysis and fluorescence recovery was made by LAS X software (Leica). Data were normalized with average value of pre-FRAP values and were plotted using Prism9 software (GraphPad), a one-phase decay fit curve was applied to the plotted results.

**Plasma membrane sheets (PM sheets) preparation**

The protocol was adapted from[54,55]. EA.hy926 cells were separated with 5 mM EDTA (ThermoFisher 15575020) on the day of cell passage and seeded on a 6-well plaque pre-coated with Poly-L-Lysine (Sigma P4707). One day after seeding, cells were rinsed twice with PBS and were submitted to sonication in 10 mL cold PBS using a probe sonicator (six pulses of 0.5 s, 20% duty cycle, output control level 2). Cytoplasm released and detached cells in the supernatant were discarded by rinsing twice with HBSS (ThermoFisher 14170088; 5 ml,

5 min). Remaining attached membranes were scrapped and resuspended in 150 µL HBSS with protease cocktail inhibitor (Sigma 4693159001) with cut pipette tips to avoid destroying the collected PM-sheets. PM-sheets were then deposited and stayed overnight into a 8-well µ-Slide (ibidi 80826) previously treated with Poly-D-Lysine (ThermoFisher A3890401) for 1 h and then coated with anti-clathrin antibodies (ThermoFisher PA5-143896) overnight. Wells were then washed twice in PBS and adherent PM-sheets were incubated in cell culture media and infected with bacteria and fixed for immunostaining.

## Image analysis

**Relative intensity accumulation quantification.** EA.hy926 cells were transfected and infected as described above. For each condition, 30 confocal images from three independent experiments of filopodia and TMS were acquired along the plasma membrane. Maximum intensity values were determined from a trace line using ImageJ software. For each image, filopodia and TMS values were normalized with the corresponding plasma membrane value and plotted using Prism9 software.

**Protein accumulation index.** The accumulation index of CD9 and Ezrin was determined by immunofluorescence volume analysis. Stack images of bacterial colony were acquired with a Zeiss Spinning Disk microscope (63X, NA 1.4). At least three experimental replicates were performed for each condition assessed, with two coverslips imaged per replicate (technical replicates). Five z-stacks (63x) were randomly taken from each coverslip. These z-stacks contained a varying number of meningococcal colonies, all of which were considered independent for the subsequent image analysis. We manually recognized bacterial colonies using Fiji ROI[64] (v2.14.0). Then with a macro designed for this work, each ROI for each image was extracted as a cropped tiff stack image. Then each cropped images channel was isolated with a Fiji macro in order to do a 3D shallow learning pixel classification with ilastik[65] (v1.4.0post1). DAPI was used to recognize bacteria. Bacteria and protein-of-interest volume (Bv and POIv, respectively) were obtained with a last Fiji macro and the accumulation index (AI) was calculated as Protein Of Interest recruitment volume divided by Bacteria volume, AI= POIv/Bv.

**Maximum tubular membrane length quantification.** The maximum tubular membrane length was assessed for $n \geq 30$ colonies among two technical replicates for a total of three experimental replicates for every experimental condition. The colonies were chosen randomly, by moving the microscope field of view and measuring each visible colony until 15×2 colonies had been quantified in the two technical replicates. Any additional colony present in the last field of view was also assessed to avoid bias. For each colony, chosen at random, CD9 immunofluorescence was used to spot tubular membrane structure. The distance between the apical plasma membrane focal plane and the tip of tubular structures was estimated with the z-stack tool on a Zeiss Apotome fluorescence microscope at x63 by multiplying the steps amount and step size of 0,24 µm.

## T4P purification

*Ammonium sulfate precipitation.* *N. meningitidis* strains were grown overnight on GCB agar plates, scraped off of the plates and resuspended in 2 mL of 20 mM ethanolamine, pH 10.5 supplemented with 1 mM dithiothreitol (DTT) at 4 °C. To shear T4P off, the bacteria were vortexed vigorously 3 times for 1 min bursts, returning them to ice for 1 min between each burst. Bacterial cells were removed from the pilus suspension by two successive centrifugations at 10,000 x g for 20 min at 4 °C. Cell pellets were resuspended in lysis buffer (50 mM Tris pH 7.5, 25 mM HEPES, 2 mM EDTA, 1% (w/v) SDS - bacterial fraction). The supernatant containing the T4P was collected and supplemented with saturated ammonium sulfate in 20 mM ethanolamine, pH 10.5 at a final concentration of 0.15 M, before overnight agitation at 4 °C. Aggregated T4P were pelleted by centrifugation at 17,000 x g for 20 min at 4 °C and resuspended overnight at 4 °C, in 400 µl of 20 mM ethanolamine, pH 10.5. The T4P solution was centrifuged at 10,000 x g for 20 min at 4 °C to remove residual cell debris and the supernatant was concentrated 10-fold using an Amicon 10 kDa MWCO membrane (Merck Millipore).

## Immunoblotting

For analysis of cell lysates, one bacterial loop of each strain of interest – grown overnight in GCB agar + corresponding antibiotics – was resuspended in 500 µl RIPA buffer (50 mM Tris pH 7,5, 150 nM NaCl, 25 nm HEPES, 2 mM EDTA, 1% w/vol SDS). This suspension was heated at 95 °C for 5 min. For analysis of T4P, pili were purified as described above. Whole lysate or T4P preparation was deposited on a 12% acrylamide SDS-PAGE gel. After transfer to nitrocellulose membrane, the membrane was incubated for 20 min in a blocking solution (PBS + 0.1% tween-20 + 4% milk) and washed twice with PBS + 0.1% tween-20 (PBST). The membrane was incubated for 1 h in the presence of anti-PilV primary antibody[10] diluted 1:1,000 in PBST, washed three times for 5 min with PBST and then incubated for 45 min with a rabbit anti-IgG secondary antibody coupled to horseradish peroxidase diluted at 1:10,000 in PBST. Proteins were detected by chemiluminescence using the Clarity Western ECL substrate detection kit (Bio-Rad) and a Chemidoc imaging system (Bio-rad). Whole proteins were detected on gel using the stain-free technology from Bio-Rad.

## Antibodies and staining reagents used in this work

FLIPPER-TR® was used at 1 µM in OptiMEM (Spirochrome AG, Switzerland) for 30mn before imaging of living infected cells. 4′,6′-diamidino-2-phenylindole (DAPI) was used at 1 µg/ml to stain eukaryotic and prokaryotic DNA. Alexa Fluor-conjugated phalloidin (Thermofisher, A22287) was used at the recommended concentration to stain filamentous actin. TS9 and TS81[66] were used at 1:200 dilution to stain for CD9 and CD81 in cytometry and microscopy assays. TS151[66] and 11B1G4 (generously provided by Leonie Ashman)[67] were used at 1:200 dilution for CD151 staining in cytometry and microscopy, respectively. Ezrin was stained with an anti-ezrin antibody at 1:500 dilution (generously provided by Dr. P. Mangeat (CNRS, UMR5539. Montpellier, France). CD44 was stained with proteintech 15675-1-AP antibody at 1:50 dilution. Alexa Fluor-conjugated goat IgG secondary antibodies (Thermofisher, A11029 and A11010) were used at 1:200 dilution in fluorescent microscopy. Clathrin polyclonal antibody (Thermofisher, PA5-143896) was used at 0.5 µg/mL in PM sheets preparation. Anti-PilV antibody[10] and HRP-conjugated goat IgG secondary antibody (Thermofisher, 656120) were used in immunoblotting at 1:10 000 dilution.

## Statistical analysis

Statistical analyses were performed with GraphPad Prism 8 or 9. For each experiment, we took the minimum number of points or pictures required to achieve a robust statistical analysis given the experimental conditions, which varied in terms of duration, microscope availability, and feasibility. When needed the data were log transformed. Normality of the distribution of whole data sets were assessed using QQ plot. The variance was assessed with a Brown-Forsythe test. When variance was different and the distribution was normal, a Brown-Forsythe and Welch ANOVA or *t*-test was performed. For non-normal distribution, a non-parametric test was performed (Mann-Whitney test or Kruskal-Wallis test). Corrections were indicated in the legend of the figures. The H0 hypothesis was rejected for a significance level of $p \leq 0.05$. Figures have been created using GraphPad Prism and Adobe Illustrator. The representative images are representative of several identical observations made in at least two independent experiments.

## Reporting summary

Further information on research design is available in the Nature Portfolio Reporting Summary linked to this article.

## Data availability

Raw data and statistical analysis are listed in the Source Data file. Source data are provided with this paper.

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

## Acknowledgements

This work was supported by the research grant ANR-19-CE14-0045-002 (to NB, PG, SM, and MC) and funding from Inserm. AL and MW were supported by a scholarship from Université Paris Cité. KS fellowship was supported by the research grant ANR-19-CE14-0045-002. We warmly thank: Nicolas Biais, Sandrine Bourdoulous, Etienne Morel, Cédric Delevoye, and Philippe Chavrier for in-depth discussions; Béatrice Durel, Julie Lesieur, Thomas Guilbert, and Pierre Bourdoncle for their expertise in microscopy (conventional, FRAP, and FLIM); Jean-Baptiste Manneville for his expertise in biophysics. Image acquisition and image analysis were performed at the Imaging Facility of Structure Fédérale de Recherche (SFR) Necker, INSERM US24/CNRS UAR3633, and the IMA-G'IC Facility of the National Infrastructure France BioImaging (ANR-10-INBS-04). Flow cytometry and analysis was performed at the Cytometry facility of SFR Necker, INSERM US24/CNRS UAR3633. Lentiviruses were obtained from the Viral Vectors and Genes Transfer facility (VVTG) platform of SFR Necker, INSERM US24/CNRS UAR3633. Illustrations were created with Biorender.com. Agreement numbers were JV28RC8O2Q for Fig. 1 and ZJ28RC80S0 for Fig. 8.

## Author contributions

Conceptualization, A.L., K.S., S.M., M.C.; Methodology, A.L., K.S., B.S., N.G., G.L., E.R., S.M., M.C.; Investigation, A.L., K.S., B.S., V.M., Y.W., M.W., M.R., T.H., L.L., MConf, J.M., P.G.; Validation, H.L., E.B., S.M., M.C.; Writing – Original Draft, A.L., K.S., S.M., M.C.; Writing – Review & Editing, A.L., K.S., B.S., N.G., A.J., N.B., P.G., E.B., G.L., E.R., S.M., M.C.; Visualization, A.L., K.S., S.M., M.C.; Funding Acquisition, N.B., S.M., M.C.; Resources, N.G., E.R.; Supervision, S.M., M.C.

## Competing interests

The authors declare no competing interests.
