## [Transparent Peer Review file · Nature Communications]

Meningococci drive host membrane tubulation to recruit their signaling receptors

Corresponding Author: Dr Mathieu Coureuil

Version 0:

Reviewer comments:

Reviewer #1

(Remarks to the Author)

This paper represents a huge amount of work indicating that pili are important for the onset of adhesion of *Neisseria* to endothelial cells and provides snapshots of the microcolonies formed upon adhesion to endothelial cells. They show that tubular cell structures form under the bacteria once adhered to the endothelial cells. These vary upon absence of different pili subunits. Furthermore, a PilT mutant and its accumulation of CD9 in fixed cells results in the conclusion that the forming of the tubules is signalling independent. These are interesting findings, and show that upon binding of a bacterium, the receiving host cell responds and reorganizes.

Comments:

The authors make quite the point of low abundance/concentration of adhesins and how this would hamper binding to receptors. However, having the pili (and Opa proteins) in the *Neisseria* OM increases local concentration. Furthermore, pili appear to function in bundles and the secondary PilV pilin/adhesin is present in multiple copies in a pilus, again increasing local concentration. There appears to be some Velcro-type of adhesion: PilC1 provides initial contact and PilV then adds to the stability of adherence with additional and multiple interactions. It is a bit unclear whether the receptors are part of the tubular membrane structures; this should be made clear.

It is clear that both PilC1 and PilV together are needed to concentrate the tubular structures under the bacteria. What is less clear from the text is what the receptor is for PilC1 (CD46 for PilC1 is under discussion). This receptor should be discussed and also whether these are thought to be part of the tubular structures.

One notices in the pictures also that the more bacteria are concentrated on the cell surface the more prominent the tubular structures appear to be. For some of the mutants tested also less bacteria seem to come together, or the surface area where the Ezrin and CD9 accumulates seems lower. Is the amount of bacteria binding together a factor and is that quantifiable?

L. 168 CD9 is more or less a proxy for other membrane proteins also found in the tubular structures. Other membrane proteins also appear to be trapped there, although not all. One misses a discussion. Why is there such a wide range of membrane proteins trapped and what do the tubular structures actually represent? Are they bundles of protein aggregates? In Fig. 2A the controls from membrane areas without bacteria also show CD9 in tubular structures but these appear longer and more flexible (FRAP), whereas in the TMS samples the structures appear more branched and broken. Do the CD9 and other proteins already localize in tubular structure, but these concentrate when bacteria are on the cell surface?

Overall, the question remains where the tubular structures under the bacteria come from. Are they present but rearranged in a different structure to accommodate *Neisseria* binding, or are they assembled upon binding. The PilT mutant and its accumulation of tubular structures on fixed cells appears to suggest re-arrangement. One wonders whether PilV and the amount of PilV proteins present on the neisserial surface would influence binding and structure formation. Is the PilV level similar in PilT mutants and WT? Are the pili extended fully when PilT is absent, or are they retracted, leading to less PilV being exposed?

minor points:

l. 79: "designed": designated (?)

l. 110; the term wetting property of the plasma membrane requires some explanation. Especially since membranes are lipidic. What is meant with the term?
l. 115: PilC1 and PilV?
l. 116: PilC1 in stead of PilV (?)
l. 175: the WT Neisseria is encapsulated. However, for additional binding the Opa proteins are used and, therefore a siaD mutant is used. This should be clarified.
On several places refs to earlier work should be added.

Reviewer #2

(Remarks to the Author)

In the current study, Mathieu Coureuil and colleagues report that *N. meningitidis* has evolved a purely physical mechanism to stimulate the formation of tubular membrane structures (TMS) at bacterial attachment sites. They provide evidence that TMS formation is independent of *N. meningitidis*-induced host signaling and is instead driven by a physical mechanism, with PilC1 being essential for initiating membrane deformation. Their data suggest that while T4P retraction promotes membrane tubulation, it does not initiate the process. Furthermore, the results show that TMS trap membrane proteins, particularly signaling receptors, while excluding unrelated ones. The authors employ a wide range of experimental approaches, including live-cell imaging, CLEM, receptor mutants, FRAP, signaling-incompetent cells, and actin inhibition. The trapping of membrane proteins (particularly signaling receptors), and the essential role of PilC1 in initiating membrane tubulation are key novel findings of this manuscript. The following points should be addressed to strengthen the study.

1. Further clarification would be valuable regarding the mechanism underlying the selective trapping of membrane proteins—such as signaling receptors (e.g., β 2AR, CD147), tetraspanins (CD9, CD81, CD151), and general surface proteins (e.g., CD4, CD44)—while excluding others like TLR2, TLR4, and the transferrin receptor. In addition, the authors could consider using unbiased proximity-based proteomic approaches (e.g., BioID) to characterize TMS-associated proteins and identify potential host interactors of PilC1
2. The authors showed that PilC1 is essential for initial adhesion and tubular membrane structure formation. In order to strengthen this finding, the authors are encouraged to assess whether complementation of the Δ C1 or Δ TC1 mutant strains with a plasmid-expressed PilC1 can rescue the phenotypes.
3. I recommend improving the clarity of the 'wetting' concept, as this term may be unfamiliar to a broader biological audience. Additionally, the authors should explicitly acknowledge and explain the previously proposed physical model of one-dimensional (1D) membrane wetting induced by T4P (Charles-Orszag et al. 2018), to help readers unfamiliar with the concept.
4. Can this physical wetting mechanism be generalized to other bacterial systems? The authors suggest so — Although not essential for this manuscript, it would be interesting to know whether the authors have considered testing this mechanism in another T4P-expressing pathogen.
5. Previous work indicated that PilC1 also triggers calcium-dependent ASM release and ceramide platform formation (Peters et al., 2019), suggesting a two-step mechanism wherein early mechanical deformation as discussed in current study, promotes receptor clustering and priming the membrane for subsequent biochemical remodelling. It would be appropriate for the authors to acknowledge and discuss this earlier work.

Reviewer #3

(Remarks to the Author)

Laurent and Sollier et. al identify pili protein from *Neisseria* that is responsible for driving a non-signaling dependent tubulation of the plasma membrane around the bacterium. The authors use multiple approaches to demonstrate that ezrin nor the actin cytoskeleton are required for the tubulation but rather show how tubulation is driven by the T4P pilus protein PilC1. While the authors show that in live cells, key *Neisseria* receptors are concentrated in these tubular structures, how PilC1 forces tubular formation in fixed cells or isolated membranes remains unclear.

Major comments

- In lines 354-355 the authors claim that “the enhanced adhesion” of the pilT mutant, but Figure 7A provides a significance value that contradicts this. The provided supplemental data also does not statistical data that demonstrates the “abolished” adhesion. This is also true of the significance claim in line 350.
- Figure 7B is not described in the main text.
- In multiple figures (3B, 3D, 4J, 4L, 5B, 5C, SF3) it is unclear why the number of points were selected for significance analysis.

Minor comments:

- On lines 115-118, “PilV” is used twice to describe two different events that are not both regulated by PilV

Version 1:

Reviewer comments:

Reviewer #3

(Remarks to the Author)

The authors have address my concerns with their additional experiments and clarifications to the text and figures.

RESPONSES TO REFEREES

We would like to thank the reviewers for their accurate reviews, positive feedback on our work, and for their comments. We have addressed all reviewers' concerns and modified the manuscript and figures in the revised version accordingly. These changes are presented point-by-point below.

Following the reviewers' comments, we have rewritten the entire introduction to make it more accessible to a broader audience.

Reviewer #1 (Remarks to the Author):

The authors make quite the point of low abundance/concentration of adhesins and how this would hamper binding to receptors. However, having the pili (and Opa proteins) in the *Neisseria* OM increases local concentration. Furthermore, pili appear to function in bundles and the secondary PilV pilin/adhesin is present in multiple copies in a pilus, again increasing local concentration.

The reviewer comment is pertinent. Because of the structure of pili, it is clear that multiple potential bacterial ligands are concentrated at their level. However, these potential ligands are tethered and cannot diffuse: they can only bind to immediately adjacent receptors, which are not quite concentrated at the host cell plasma membrane. In comparison, the concentration of soluble ligand is globally higher and these ligands are free to diffuse. It is precisely the wetting process, which "brings" concentrated receptors close to the ligands thus facilitating receptor binding and activation.

There appears to be some Velcro-type of adhesion: PilC1 provides initial contact and PilV then adds to the stability of adherence with additional and multiple interactions.

Correct, this is our overall conclusion

It is a bit unclear whether the receptors are part of the tubular membrane structures; this should be made clear.

Not only the specific receptors are parts of these tubular membrane structures, their density is also enhanced in these structures compared to their basal density at the plasma membrane. We tried to make it clearer in the revised version of the manuscript. For example, see Figure 2 and lines 128-134 in the text.

It is clear that both PilC1 and PilV together are needed to concentrate the tubular structures under the bacteria. What is less clear from the text is what the receptor is for PilC1 (CD46 for PilC1 is under discussion). This receptor should be discussed and also whether these are thought to be part of the tubular structures.

At the moment the receptor for PilC1 remains unknown. The role of CD46 is still debated in the *Neisseria* community. We assume that, due to the low number of PilC molecules at the tip of T4P, the PilC-interacting "factor" is abundantly expressed at the plasma membrane of host cells. Also based on our previous findings showing that meningococcal ligands interact with CD147 and β 2AR glycans, our hypothesis is that this unknown PilC-interacting "factor" might also be a glycan. We are currently investigating this issue.

In the revised version of the manuscript we mentioned the fact that PilC1 interacts "an unknown receptor" line 136 and discussed the issue lines 410-417 in the discussion section.

One notices in the pictures also that the more bacteria are concentrated on the cell surface the more prominent the tubular structures appear to be. For some of the mutants tested also less bacteria seem

to come together, or the surface area where the Ezrin and CD9 accumulates seems lower. Is the amount of bacteria binding together a factor and is that quantifiable?

Indeed, the greater the number of bacteria, the greater the recruitment of CD9 or ezrin. However, our quantification strategy in Figures 3B, 3D, 4J, 4L, 5B and 5C takes this point into account since the recruitment of CD9 or ezrin is normalised to the surface of bacteria. Therefore, in Figure 3, despite the presence of fewer bacteria after 2 hours of NSC66 treatment, there is no decrease in normalised CD9 labelling. This issue better explained in the revised version of the manuscript and in the methods section.

CD9 is more or less a proxy for other membrane proteins also found in the tubular structures. Other membrane proteins also appear to be trapped there, although not all. One misses a discussion. Why is there such a wide range of membrane proteins trapped and what do the tubular structures actually represent? Are they bundles of protein aggregates?

This is an interesting question for which we do not have answers yet. We mentioned in the discussion section starting line 430: *“Our results (see Figure 2F) also suggest an active receptor accumulation phenomenon in living cells that has yet to be determined. Moreover, it is not clear how other integral membrane proteins, such as the transferrin receptor or Toll like receptors are excluded from these structures.”*

Based on our results and those of other teams, we currently have no argument to support or refute the possibility that proteins in tubular membrane structures (TMS) are aggregated. It is clear, however that the β 2AR maintains its signalling properties in TMS.

In Fig. 2A the controls from membrane areas without bacteria also show CD9 in tubular structures but these appear longer and more flexible (FRAP), whereas in the TMS samples the structures appear more branched and broken.

We assume that reviewer #1 is referring to the CD9 labelling in the top panel of Figure 2A. If correct, we think that our wording in the legend was misleading. Figure 2A top panel actually shows plasma membrane labelling at the cell periphery, not in tubular structures. We have amended the text of the legend accordingly.

Do the CD9 and other proteins already localize in tubular structure, but these concentrate when bacteria are on the cell surface?

Overall, the question remains where the tubular structures under the bacteria come from. Are they present, but rearranged, in a different structure to accommodate Neisseria binding, or are they assembled upon binding.

CD9 is known to be distributed all over the plasma membrane and enriched at cell-cell junctions, at the front of migration and in membrane curvatures such as filopodia. However, we do not detect tubular structures at the apical plasma membrane in non-infected endothelial cells. We have added this information in the supplementary Figure 1C and lines 170-172. Consequently, we assume that TMS are formed upon binding.

The PilT mutant and its accumulation of tubular structures on fixed cells appears to suggest re-arrangement. One wonders whether PilV and the amount of PilV proteins present on the neisserial surface would influence binding and structure formation. Is the PilV level similar in PilT mutants and WT? Are the pili extended fully when PilT is absent, or are they retracted, leading to less PilV being exposed?

Figure 7B left panel shows that PilV is expressed at the same level in the T4P of wild type and $\Delta TC1$ bacteria (devoid of PilT and PilC1). Thus, PilV molecules are still exposed in the $\Delta TC1$ mutant. PilT is responsible for retraction while PilF is necessary for elongation. Bacteria devoid of PilT express long and immobile T4P. This issue is now explained lines 264-265.

minor points:

I. 79: “designed”: designated (?)

Corrected

I. 110; the term wetting property of the plasma membrane requires some explanation. Especially since membranes are lipidic. What is meant with the term?

We agree with the reviewer that this information was missing. The term wetting is now explained lines 122-127. *“Wetting is a mechanical phenomenon describing the capacity of a liquid to spread across a solid surface. Expansion of the contact interface is driven by diverse molecular interactions, including ionic, electrostatic, polar, and van der Waals forces, and proceeds as long as the process remains energetically favorable. This principle underlies phenomena such as capillary action. In the one-dimensional (1D) wetting model proposed by Charles-Orszag and colleagues, the plasma membrane behaves as a fluid, spreading along adherent nanofibers.”*

I. 115: PilC1 and PilV?

Corrected

I. 116: PilC1 in stead of PilV (?)

Corrected

I. 175: the WT Neisseria is encapsulated. However, for additional binding the Opa proteins are used and, therefore a siaD mutant is used. This should be clarified.

We have now explained the experimental design in lines 195-198.

On several places refs to earlier work should be added.

We have added several references throughout the introduction section.

Reviewer #2 (Remarks to the Author):

1. Further clarification would be valuable regarding the mechanism underlying the selective trapping of membrane proteins—such as signaling receptors (e.g., $\beta 2AR$, CD147), tetraspanins (CD9, CD81, CD151), and general surface proteins (e.g., CD4, CD44)—while excluding others like TLR2, TLR4, and the transferrin receptor.

Reviewer #2 raises an important question. It appears that all tested receptors are accumulated in TMS with the exception of transferrin receptor and TLR4/2. As mentioned in the discussion section, it is not clear how these membrane proteins are excluded (or not accumulated) from these structures. From the discussions with the biophysicists involved in our study (Nicolas BORGHI, Philippe GIRARD - both authors - or Jean-Baptiste MANNEVILLE - acknowledged), no readily testable hypotheses have emerged. We feel that demonstrating the mechanism of exclusion or non-inclusion of TfR and TLR is beyond the scope of the present work.

In addition, the authors could consider using unbiased proximity-based proteomic approaches (e.g., BioID) to characterize TMS-associated proteins and identify potential host interactors of PilC1

The comment of reviewer #2 actually raises two questions.

(1) What are the host cell interactors of PilC1?

The same question was raised by reviewer #1. At the moment, the receptor for PilC1 remains unknown. We assume that, due to the low number of PilC molecules at the tip of T4P, the PilC-interacting “factor” is abundantly expressed at the plasma membrane of host cells. Also, based on our previous findings showing that meningococcal ligands interact with CD147 and β 2AR glycans, our hypothesis is that this unknown PilC-interacting “factor” might also be a glycan. We are currently investigating this issue (see discussion section, lines 410-417). Using recombinant PilC or a PilC domain to fish-up interacting partners is a possible option, although PilC is insoluble and impossible to purify so far by researcher in the field.

(2) What are the TMS-associated proteins?

A turboID screen using the recombinant protein CD9-TurboID in infected epithelial cells has been deposited on BiorXiv by a competitor group (<https://doi.org/10.1101/2024.12.13.628358>). They identified CD147, CD44, etc. This strategy is appealing, but generated a massive number of potential targets likely related to the multiple properties of CD9; determining which ones might selectively be involved in the interaction with PilC will be quite challenging. Repeating a similar experiment on endothelial cells would be very time-consuming because of the numerous technical challenges. It would represent a new project on its own.

2.The authors showed that PilC1 is essential for initial adhesion and tubular membrane structure formation. In order to strengthen this finding, the authors are encouraged to assess whether complementation of the Δ C1 or Δ TC1 mutant strains with a plasmid-expressed PilC1 can rescue the phenotypes.

We have included the requested complementation experiment in the new figure 7 and its legend.

3.I recommend improving the clarity of the 'wetting' concept, as this term may be unfamiliar to a broader biological audience. Additionally, the authors should explicitly acknowledge and explain the previously proposed physical model of one-dimensional (1D) membrane wetting induced by T4P (Charles-Orszag et al. 2018), to help readers unfamiliar with the concept.

We agree with the reviewer that this information was missing. The term wetting is now explained lines 122-127. *“Wetting is a mechanical phenomenon describing the capacity of a liquid to spread across a solid surface. Expansion of the contact interface is driven by diverse molecular interactions, including ionic, electrostatic, polar, and van der Waals forces, and proceeds as long as the process remains energetically favorable. This principle underlies phenomena such as capillary action. In the one-dimensional (1D) wetting model proposed by Charles-Orszag and colleagues, the plasma membrane behaves as a fluid, spreading along adherent nanofibers.”*

4.Can this physical wetting mechanism be generalized to other bacterial systems? The authors suggest so — Although not essential for this manuscript, it would be interesting to know whether the authors have considered testing this mechanism in another T4P-expressing pathogen.

We believe that this wetting mechanism should be generalized to other systems, first in pilated bacteria and then in mammalian cells, as mentioned lines 44-455 in the discussion section. So far we could observe the same phenomenon with the close relative *N. gonorrhoeae*, which possesses the same T4P machinery with similar PilC and PilV proteins.

5.Previous work indicated that PilC1 also triggers calcium-dependent ASM release and ceramide platform formation (Peters et al., 2019), suggesting a two-step mechanism wherein early mechanical

deformation as discussed in current study, promotes receptor clustering and priming the membrane for subsequent biochemical remodelling. It would be appropriate for the authors to acknowledge and discuss this earlier work.

We overlooked this important point. We have added a sentence in the discussion section line 426-429.

Reviewer #3 (Remarks to the Author):

While the authors show that in live cells, key *Neisseria* receptors are concentrated in these tubular structures, how PilC1 forces tubular formation in fixed cells or isolated membranes remains unclear.

We have added new confocal microscopy pictures confirming the accumulation of β 2AR and CD4 after infection of fixed cells (new Figure S5). The PilC1 role in this context is to anchor the pilus to the plasma membrane, which is permissive for wetting, the physical phenomenon leading to β 2AR accumulation in TMS. Wetting, which does not require cell signalling (paragraph starting line 287) or ATP production (paragraph starting line 320), can indeed occur in fixed cells or isolated membranes.

Major comments

- In lines 354-355 the authors claim that “the enhanced adhesion” of the *pilT* mutant, but Figure 7A provides a significance value that contradicts this. The provided supplemental data also does not statistical data that demonstrates the “abolished” adhesion.

This is also true of the significance claim in line 350.

Following #Reviewer 2 request, we have included a complementation experiment in the new figure 7 and its legend. All experimental points of the new figure have been generated in a repeated independent study (explaining the different distribution of values), which provided similar results. The sentence pointed by #Reviewer 3 specifically referred to the enhanced adhesion observed in fixed cells infected by the Δ *pilT* strain ($p < 0.0001$ in the new figure). No significant enhancement in adhesion was found for the bacterial strain missing PilT on living cells ($p = 0.780$ in the new figure). We modified the sentence to avoid any potential confusion (lines 374-376)

The words “significantly inhibited adhesion” line 371 referred to the “genetic deletion of PilC1” line 349.

- Figure 7B is not described in the main text.

This omission was corrected in the revised version.

- In multiple figures (3B, 3D, 4J, 4L, 5B, 5C, SF3) it is unclear why the number of points were selected for significance analysis.

We have now added sentences in the statistical analysis section to explain this specific point in the “Methods” section. In the Statistical analysis paragraph: “For each experiment, we took the minimum number of points or pictures required to achieve a robust statistical analysis given the experimental conditions, which varied in terms of duration, microscope availability, feasibility”; In the Image analysis paragraph: “At least three experimental replicates were performed for each condition assessed, with two coverslips imaged per replicate (technical replicates). Five z-stacks (63x) were randomly taken from each coverslip. These z-stacks contained a varying number of meningococcal colonies, all of which were considered independent for the subsequent image analysis.” and “The colonies were chosen randomly, by moving the microscope field of view and measuring each visible colony until 15x2 colonies had been quantified in the two technical replicates. Any additional colony present in the last field of view was also assessed to avoid bias.”.

Minor comments:

- On lines 115-118, "PiIV" is used twice to describe two different events that are not both regulated by PiIV

We corrected the mistake: PiIC1 and PiIV are both needed. See line 135.